# Personalized Federated Learning
# via Low-Rank Matrix Optimization

**Ali Dadras**                                                                                    *ali.dadras@umu.se*
*Umeå University, Sweden*

**Sebastian U. Stich**                                                                                  *stich@cispa.de*
*CISPA Helmholtz Center, Germany*

**Alp Yurtsever**                                                                              *alp.yurtsever@umu.se*
*Umeå University, Sweden*

**Reviewed on OpenReview:** *https://openreview.net/forum?id=DFJu1QB2Nr*

## Abstract

Personalized Federated Learning (pFL) has gained significant attention for building a suite of models tailored to different clients. In pFL, the challenge lies in balancing the reliance on local datasets, which may lack representativeness, against the diversity of other clients' models, whose quality and relevance are uncertain. Focusing on the clustered FL scenario, where devices are grouped based on similarities in their data distributions without prior knowledge of cluster memberships, we develop a mathematical model for pFL using low-rank matrix optimization. Building on this formulation, we propose a pFL approach leveraging the Burer-Monteiro factorization technique. We examine the convergence guarantees of the proposed method and present numerical experiments on training deep neural networks, demonstrating the empirical performance of the proposed method in scenarios where personalization is crucial.

## 1 Introduction

Federated Learning (FL) is an important paradigm in machine learning, holding a great promise for training machine learning models over a large network with restricted data sharing. It is most suitable when clients require collaboration—often due to the absence of a large, representative dataset available at hand that can capture the diversity and variability of the underlying behavior—but in an environment where sharing datasets with collaborators is prohibited—often driven by concerns and regulations surrounding data sharing and storage. Consequently, FL research has been focused on designing algorithms that can solve optimization and learning problems on a network without sharing essential data; but instead communicating decision variables (referred to as local models in machine learning) or other auxiliary variables like gradients or step-directions, alongside privacy improvement strategies such as encryption or noise injection. However, the restriction of data sharing presents challenges beyond the training process. Primarily, limitations on data sharing hinder effective control over the quality and relevance of the data provided by participating clients—a major concern that led to the rise of personalized federated learning models (pFL).

The primary goal of pFL is finding the right balance between the two conflicting forces: the reliability of local datasets which may lack representativeness, and the diversity of collaborators' models whose quality and relevance are uncertain. As a result, pFL lacks a clear definition and direction without explicit specifications regarding data distribution of the clients. Regrettably, it appears there is no universally accepted measurable quantitative goal to evaluate the success in personalization. Adding to this concern, many existing pFL methods are tested in settings that are inherently unsuited for pFL, where either Federated Averaging (FedAvg) or local training produces the best accuracies. It is evident that when FedAvg yields optimal results,

indicating uniform data distributions across clients, there is no room for personalization. On the other hand, if local training performs well, suggesting that local datasets represent the problem sufficiently well, the necessity of FL is called into question.

Motivated by these observations, our first step is to formulate a mathematical problem that highlights the role and necessity of a pFL approach. Suppose there are $n$ clients collaborating on an FL system, indexed by $i = 1, \ldots, n$, and assume that the data for each client comes from a specific data distribution, denoted by $\mathcal{D}_i$. The true objective function for each client is defined as:

$$f_i^{\natural}(\boldsymbol{\theta}_i) := \mathbb{E}_{\boldsymbol{\xi}_i \sim \mathcal{D}_i} \, \ell_i(\boldsymbol{\theta}_i, \boldsymbol{\xi}_i), \tag{1}$$

where $\ell_i : \mathbb{R}^d \times \mathbb{R}^p \to \mathbb{R}$ is a loss function. Here, $\boldsymbol{\xi}_i = (\boldsymbol{x}_i, \boldsymbol{y}_i)$ denotes a random data sample drawn from the probability distribution $\mathcal{D}_i$, where $\boldsymbol{x}_i$ represents the input features and $\boldsymbol{y}_i$ denotes the corresponding response. When data distributions are known, a solution to this problem can be found by minimizing $f_i^{\natural}(\boldsymbol{\theta}_i)$ locally:

$$\min_{\boldsymbol{\theta}_i} \; f_i^{\natural}(\boldsymbol{\theta}_i). \tag{2}$$

However, the true distribution $\mathcal{D}_i$ is unknown in practice. Instead, client $i$ approximates its objective using an empirical dataset $\mathcal{M}_i = \{\boldsymbol{\xi}_{i,1}, \ldots, \boldsymbol{\xi}_{i,m_i}\}$ consisting of $m_i$ independent and identically distributed data samples drawn from $\mathcal{D}_i$. This yields the empirical objective

$$f_i(\boldsymbol{\theta}_i) := \frac{1}{m_i} \sum_{\boldsymbol{\xi}_i \in \mathcal{M}_i} \ell_i(\boldsymbol{\theta}_i, \boldsymbol{\xi}_i) \,.$$

Throughout, we operate under the assumption that the dataset $\mathcal{M}_i$ is not large enough for clients to accurately approximate a solution to problem (2) locally on their own. Otherwise, FL would not be required.

An effective solution to this problem is possible only if the distributions $\mathcal{D}_i$ exhibit some correlation that we can exploit. At one extreme, when all distributions are the same, the standard FL template can be used, which can be formulated as

$$\min_{\boldsymbol{\theta}_1, \ldots, \boldsymbol{\theta}_n} \; \frac{1}{n} \sum_{i=1}^{n} f_i(\boldsymbol{\theta}_i) \quad \text{s.t.} \quad \boldsymbol{\theta}_1 = \cdots = \boldsymbol{\theta}_n \,, \tag{3}$$

or equivalently as

$$\min_{\boldsymbol{\theta}} \; \frac{1}{n} \sum_{i=1}^{n} f_i(\boldsymbol{\theta}) \,. \tag{4}$$

A significant portion of existing pFL methods are designed by relaxing the equality constraint (also called consensus constraint); examples include Moreau envelope smoothing and quadratic penalty regularization (Dinh et al., 2020; Li et al., 2020). However, these approaches that penalize model dissimilarity using a specific norm have limitations, as they rely on the (implicit) assumption that similarity in distributions $\mathcal{D}_i$ translates to the proximity of client models in a given norm. The following simple examples demonstrate these limitations:

**Example 1** (Label noise in classification). *Consider a linear binary classification problem with two groups of clients that differ in their sign conventions. Specifically, these groups label the positive and negative classes in opposite ways due to a misalignment in how they interpret the binary outcomes. Clearly, the data distributions of these two groups are nearly identical, differing only by one bit. However, for this linear classifier, this difference results in the optimal models for the two groups having opposite signs, leading to solutions $\boldsymbol{\theta}_{group1}^{\star} = -\boldsymbol{\theta}_{group2}^{\star}$, which are distant in all norms.*

Although presented as a toy example here, mislabeling is a common problem in classification tasks, especially in domains like healthcare where human experts are involved in data collection. In real-world FL systems, where data remains private, detecting or preventing such issues is challenging. Therefore, algorithms must be designed robust against these inconsistencies.

**Example 2** (Clustered FL). *Suppose each client draws data from one of $r$ distinct distributions, forming $r$ clusters of clients. We assume that cluster memberships are unknown, and the challenge is to establish effective collaboration without knowing in advance which clients share similar data distributions.*

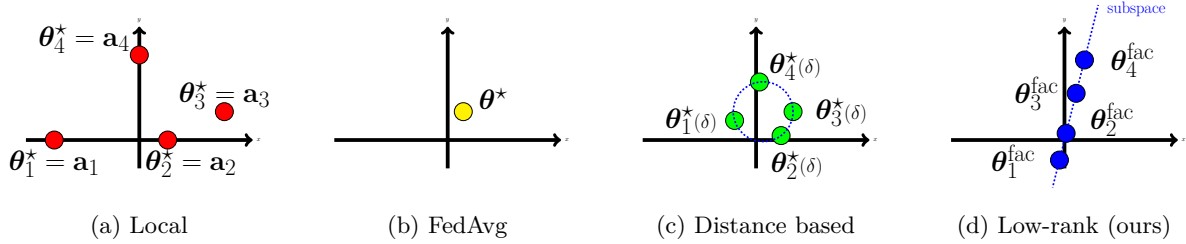

Figure 1: Solutions to the minimization problem $\frac{1}{n}\sum_{i=1}^{n} \|\boldsymbol{\theta}_i - \mathbf{a}_i\|^2$, where $\boldsymbol{\theta}_i, \mathbf{a}_i \in \mathbb{R}^2$. The red points represent individual minimizers of the problem which are equivalent to $\boldsymbol{\theta}_i^\star = \mathbf{a}_i$. FEDAVG solution is $\boldsymbol{\theta}^\star = \frac{1}{n}\sum_{i=1}^{n} \mathbf{a}_i$ which is shown in yellow. Considering $\|\boldsymbol{\theta}_i - \boldsymbol{\theta}_j\| \leq \delta$ constraint one can find $\boldsymbol{\theta}_i^\star(\delta)$ as shown in green. Solving this problem with the low-rank assumption ($r = 1$) gives us solutions $\boldsymbol{\theta}_i^{\text{fac}}$, blue points, lying in a low-dimensional subspace.

A specific class of pFL methods for clustered FL problems focuses on identifying clusters using various similarity measures and then performing cluster-aware aggregation. However, in a real-world FL system where data remains private, estimating these clusters is challenging without exposing additional information. This difficulty persists unless we rely on the stringent assumption that similar distributions produce models that are close in a certain norm, which could then be used to estimate clusters during training.

**Example 3** (Collaborative filtering). *Consider the classical problem of recommendation systems. Suppose there are $n$ clients and $p$ items. Let $\boldsymbol{\theta}_i \in \mathbb{R}^p$ represent the relevance scores of client $i$ for the items. The data consists of the actual scores rated by the clients, where each client rates only a subset of the items, denoted by $S_i \subseteq \{1, \ldots, p\}$. We denote these scores by $\boldsymbol{\theta}_{ij}^\star$ for $j \in S_i$. The goal is predicting unknown scores that are not in $S_i$, based on hidden patterns among clients.*[1]

Models based on Euclidean distance regularizations are known to fail in accurately predicting movie preferences. Instead, low-rank matrix factorization is among the most popular and successful approaches to collaborative filtering, as demonstrated by the Netflix competition 2008 progress prize winning team (Koren et al., 2009). The typical explanation for the empirical success of low-rank models in this problem is that movie preferences are well-parameterized by a few meaningful factors, such as genre, cast, language, and year. A more nuanced argument generalizes this by noting that low-rank matrices naturally arise in latent variable models (LVMs). While this is standard for LVMs with linear parameterizations, (Udell & Townsend, 2019) demonstrate that low-rank models are effective for a broad class of (possibly high-dimensional) LVMs parameterized by a piecewise analytic function.

Inspired by these examples, we explore how to formulate pFL without relying on a specific distance metric. This leads us to investigate low-dimensional subspace formulations, where personalized models are related by their membership to a low-dimensional subspace rather than their proximity in a distance metric. This approach allows us to conceptualize pFL by focusing on the inherent structure of the model relationships rather than their spatial closeness, as illustrated in Figure 1. Drawing parallels to collaborative filtering, we specifically focus on low-rank formulations.

We can now summarize our main contributions:

- We introduce a new formulation for pFL based on low-rank matrix optimization in Section 3. Utilizing a nonconvex matrix factorization method applied to this formulation, we propose a new method called Personalized Federated Learning via Matrix Factorization (pFL$^{\text{MF}}$).

- We investigate the convergence guarantees of the proposed method in Section 4. For the smooth nonconvex minimization problem, we show that the proposed method converges to a first-order stationary point at a rate of $\mathcal{O}(1/T)$; with the stochastic gradients, the rate becomes $\mathcal{O}(1/\sqrt{T})$.

---

[1]The decision variable in matrix completion reveals the data, limiting the privacy benefits of FL. Nevertheless, the problem highlights the challenge of distributed learning with personalized models.

- We present numerical experiments on training various types of neural networks in Section 5. We compare the performance of the proposed method against the baseline in scenarios where personalization is crucial, such as classification in clustered FL with label misalignment or non-homogeneous data distributions.

## 2 Related Work

Many pFL algorithms regularize client models to remain close; the main assumption is that personal models are close under a chosen metric (Dinh et al., 2020). Learning a mixture of the global model and the local models is proposed in (Hanzely & Richtárik, 2020), where these personalized models are encouraged to stay close to their average by incorporating a quadratic penalty. Empirical evidence suggests that model similarities among different neural networks, particularly in their classifier layers, correlates with the similarity of the training data distributions (Tan et al., 2023). Thus, enforcing model closeness (e.g., in Euclidean distance) tends to favor clients with similar data distributions.

More recently, model decoupling methods have been proposed (Arivazhagan et al., 2019; Oh et al., 2021; Pillutla et al., 2022; Mishchenko et al., 2023), showing a better performance than distance-regularization-based pFL methods. The main idea is to decouple each local model into two blocks, a feature extractor block followed by a classifier block. The feature extractor block is communicated and aggregated over clients and the classifier block is trained locally by each client. (Arivazhagan et al., 2019) considered a personalization of only selected layers of the neural network: all devices share a common set of base layers with identical weights, while each device maintains its own personalization layers that adapt to its local data. The base layers are synchronized with the server while the personalization layers are kept private by each device. In (Oh et al., 2021), the entire network is decomposed into the body (extractor), which is related to universality, and the head (classifier), which is related to personalization. This reduces the update and aggregation parts from the entire model to the body of the model during federated training.

Perhaps the most relevant works to ours are (Collins et al., 2021) and (Thekumparampil et al., 2021). The goal in (Thekumparampil et al., 2021) is to find a shared low-dimensional representation of the data features. While the primary focus of (Thekumparampil et al., 2021) is on multi-task learning, it can be applied for personalized FL by treating each client's learning problem as a separate task, as considered in (Collins et al., 2021). In (Collins et al., 2021), the server learns the common low-dimensional features of the data, and each client learns local features suited to its requirements. This method, Federated Representation Learning (FedRep), uses gradient-based updates to train a global low-dimensional representation and allows each client to compute a personalized low-dimensional classifier. The key difference between FedRep and our approach is in the low-rank assumption. FedRep assumes that each clients parameters are low rank. In contrast, we assume that the concatenation of all clients parameters lies in a low-dimensional space. In other words, their feature extractor extracts the features from a single shared global model while our method trains a set of models and each client uses a combination of these models as its personalized model. Put differently, FedRep assumes that $\boldsymbol{\theta}_i$ are low rank, whereas we assume that $\boldsymbol{\Theta} := [\text{vec}(\boldsymbol{\theta}_1), \text{vec}(\boldsymbol{\theta}_2), \ldots, \text{vec}(\boldsymbol{\theta}_n)]$ is low rank. Furthermore, FedRep focuses on the linear representation setting with a quadratic loss.

Beyond distance-based and decoupling approaches, a variety of other strategies have been explored for pFL. (Zhang et al., 2024) introduces LR-BPFL, a Bayesian personalized federated learning method that learns a global deterministic model along with personalized low-rank Bayesian corrections. (Bao et al., 2023) proposes FEDCOLLAB, a clustered federated learning framework that mitigates negative transfer by partitioning clients into non-overlapping coalitions informed by both pairwise distribution distances and relative data quantities. (Prakash et al., 2023) processes hierarchical, tree-like data in federated learning by developing an algorithm tailored to hyperbolic spaces. (Anelli et al., 2022) investigates federated pair-wise learning for factorization models in a recommendation scenario. (Huang et al., 2022) proposes an FL framework for solving the Point-of-Interest (POI) recommendation problem. (Ammad-Ud-Din et al., 2019) introduces a federated implementation of collaborative filtering for recommendation systems. Liang et al. (2020) introduces LG-FEDAVG, which combines local representation learning with global model learning in an end-to-end manner. Each local device learns to extract higher-level representations from raw data before a global model operates on the representations (rather than raw data) from all devices. (Tan et al., 2023) proposes a decoupling algorithm that also personalizes feature extractors by adjusting aggregation weights based on

classifier similarity. (Deng et al., 2020) introduces APFL algorithm, which learns a personalized model for each user as a convex combination of local and global models, with the combination coefficients adaptively updated during training. (Hao et al., 2022) assumes factorized weights for neural network weights and, rather than learning a single global model, learns a dictionary of rank-1 weight factor matrices. Each client then assembles a personalized model from this dictionary to match its own data distribution. (Jeong & Hwang, 2022) considers factorization of the model parameters and allows clients to perform a selective aggregation scheme to utilize only the knowledge from the relevant participants for each client. (Pal et al., 2024) introduces LRS, modeling user parameters as low-rank + sparse to capture shared structure and individual-specific characteristics. They further develop AMHT-LRS, providing theoretical guarantees in the linear Gaussian setting, and extend it to a user-level differentially private version.

Finally, several works use low-rank matrix optimization in FL, but with different motivations and goals from ours. One line uses low-rank structure in the model parameters: (Pinto et al., 2023) projects private datasets onto a low-dimensional subspace spanned by principal components estimated from public unlabeled data, then apply gradient-based private algorithms (e.g., Noisy-SGD) to learn a linear classifier; (Zhao et al., 2016; Cai et al., 2014) assumes one of the neural network layers is low rank; (Liu et al., 2024a) uses homogeneous pre-factorized low-rank layers across clients; (Tran et al., 2025) factorizes local prompts into two lower-rank components plus a residual; and (Niu et al., 2023) introduces PriSM training, which assigns resource-constrained clients low-rank sub-models via importance-aware probabilistic sampling. A second line uses low-rank structure in the gradients, such as (Kasiviswanathan, 2021; Yu et al., 2021; Gooneratne et al., 2020). (Yao et al., 2021) proposes FedHM, which trains low-rank factorized neural networks of a specified size and reconstructs a full-rank global model on the server via a model shape alignment method.

## 3  Algorithm

We propose a novel formulation for pFL based on low-rank matrix optimization:

$$\min_{\boldsymbol{\Theta} \in \mathbb{R}^{d \times n}} F(\boldsymbol{\Theta}) := \frac{1}{n} \sum_{i=1}^{n} f_i(\boldsymbol{\theta}_i) \quad \text{s.t.} \quad \text{rank}(\boldsymbol{\Theta}) \leq r. \tag{5}$$

Here, $\boldsymbol{\Theta} \in \mathbb{R}^{d \times n}$ denotes the system-level decision variable obtained by concatenating clients' decision variables as $\boldsymbol{\Theta} := [\boldsymbol{\theta}_1, \boldsymbol{\theta}_2, \ldots, \boldsymbol{\theta}_n]$, and $r$ is problem specific tuning parameter. Note that this formulation suits well for the examples we discussed in the introduction.

There exists a rich literature on rank-constrained matrix optimization problems, approaches including hard thresholding algorithms (Jain et al., 2010; Goldfarb & Ma, 2011; Kyrillidis & Cevher, 2014), convex relaxation methods (Candès & Recht, 2012; Recht et al., 2010), and nonconvex matrix factorization techniques (Burer & Monteiro, 2003; Sun & Luo, 2016; Bhojanapalli et al., 2016; Park et al., 2017). The first two class of algorithms require expensive spectral decomposition steps, hence they are not suitable for federated implementation unless the server possesses sufficient computational power to perform such decompositions. Consequently, we adopt the nonconvex matrix factorization technique, also known as the Burer-Monteiro (BM) factorization.

BM factorization strategy replaces the system-level decision variable $\boldsymbol{\Theta} \in \mathbb{R}^{d \times n}$ with a factorized form of $\boldsymbol{\Theta} = \mathbf{U}\mathbf{V}^\top$. This transformation leads to the following optimization problem:

$$\min_{\mathbf{U} \in \mathbb{R}^{d \times r}, \mathbf{V} \in \mathbb{R}^{n \times r}} \psi(\mathbf{U}, \mathbf{V}), \quad \text{where} \quad \psi(\mathbf{U}, \mathbf{V}) := F(\mathbf{U}\mathbf{V}^\top) = \frac{1}{n} \sum_{i=1}^{n} f_i(\mathbf{U}\mathbf{v}_i). \tag{6}$$

We denote by $\mathbf{V}^\top := [\mathbf{v}_1, \cdots, \mathbf{v}_n] \in \mathbb{R}^{r \times n}$. In this notation, personalized model parameters can be computed as $\boldsymbol{\theta}_i = \mathbf{U}\mathbf{v}_i \in \mathbb{R}^d$. One can interpret $\mathbf{U} \in \mathbb{R}^{d \times r}$ as a shared feature representation in the FL problem, computed by the server, and $\mathbf{v}_i \in \mathbb{R}^r$ as the feature extractor specific to client $i$.

While various optimization techniques can address problem (6), we design our algorithm based on the simple block-coordinate gradient updates. We can compute the gradient of $\psi$ with respect to $\mathbf{U}$ and $\mathbf{v}_i$ as follows:

$$\nabla_{\mathbf{U}} \psi(\mathbf{U}, \mathbf{V}) = \frac{1}{n} \sum_{i=1}^{n} \nabla f_i(\mathbf{U}\mathbf{v}_i) \, \mathbf{v}_i^\top \quad \text{and} \quad \nabla_{\mathbf{v}_i} \psi(\mathbf{U}, \mathbf{V}) = \frac{1}{n} \mathbf{U}^\top \nabla f_i(\mathbf{U}\mathbf{v}_i). \tag{7}$$

---

**Algorithm 1** Personalized Federated Learning via Matrix Factorization (`pFL`$^{\text{MF}}$)

---

$\quad$ **set** $\mathbf{U}^0 \in \mathbb{R}^{m \times r}, \mathbf{v}_i^0 \in \mathbb{R}^r \ \forall i \in [n]$.

$\quad$ **for** round $t = 0, 1, \ldots, T-1$ **do**

$\qquad$ — **Client**-level local training ————————-

$\qquad$ **for** client $i \in \mathcal{S}_t$ **do**

$\qquad\quad$ set $\mathbf{v}_i^{t,1} = \mathbf{v}_i^t$.

$\qquad\quad$ **for** $k = 0, \ldots, K-1$ **do**

$\qquad\qquad$ $\mathbf{v}_i^{t,k+1} = \mathbf{v}_i^{t,k} - \eta_i \frac{1}{n} \mathbf{U}^{t\top} \ \nabla f_i(\mathbf{U}^t \mathbf{v}_i^{t,k})$

$\qquad\quad$ **end for**

$\qquad\quad$ $\mathbf{v}_i^{t+1} = \mathbf{v}_i^{t,K}$

$\qquad\quad$ $\mathbf{G}_i^t = \left( \nabla f_i(\mathbf{U}^t \mathbf{v}_i^t) \right) \mathbf{v}_i^{t\top}$

$\qquad\quad$ Client communicates $\mathbf{G}_i^t$ to the server.

$\qquad$ **end for**

$\qquad$ — **Server**-level aggregation ————————

$\qquad$ $\mathbf{U}^{t+1} = \mathbf{U}^t - \eta_i \frac{1}{|\mathcal{S}_t|} \sum_{i \in \mathcal{S}_t} \ \mathbf{G}_i^t$

$\qquad$ Server communicates $\mathbf{U}^{t+1}$ to the clients.

$\quad$ **end for**

---

It is crucial that $\psi$ is separable with respect to $\mathbf{v}_i$, enabling clients to compute $\nabla_{\mathbf{v}_i} \psi(\mathbf{U}, \mathbf{V})$ in parallel without requiring access to data or model parameters from other clients, given the features $\mathbf{U}$. Consequently, for a given step-size $\eta_i > 0$, local training steps can be independently formulated and performed by each participating client as:

$$\mathbf{v}_i^{t+1} = \mathbf{v}_i^t - \eta_i \frac{1}{n} \mathbf{U}^{t\top} \ \nabla f_i(\mathbf{U}^t \mathbf{v}_i^t). \tag{8}$$

On the other hand, $\psi$ is not separable with respect to the rows or columns of $\mathbf{U}$, necessitating collaboration among clients for computing $\nabla_{\mathbf{U}} \psi(\mathbf{U}, \mathbf{V})$. Consequently, the gradient step in $\mathbf{U}$ requires communication and will be performed at the server, forming our aggregation step:

$$\mathbf{U}^{t+1} = \mathbf{U}^t - \frac{1}{n} \sum_{i=1}^n \ \eta_i \left( \nabla f_i(\mathbf{U}^t \mathbf{v}_i^t) \right) \mathbf{v}_i^{t\top}. \tag{9}$$

Algorithm 1 depicts the pseudo-code of our algorithm. Here, $K$ is the number of local passes each client performs, and the output of the algorithm is a set of personalized parameters $\boldsymbol{\theta}_i = \mathbf{U} \mathbf{v}_i$ that each client can compute locally using its feature extractors $\mathbf{v}_i$ and the shared feature representation $\mathbf{U}$.

## 4 Convergence Guarantees

Several works have studied the convergence for the problem (6) under different assumptions; we refer to (Chi et al., 2019) and references therein. (Bhojanapalli et al., 2016; Park et al., 2018) proved linear/sub-linear rates for smooth functions and smooth and strongly convex functions, respectively. Due to the nonconvex nature of BM factorization, even in cases where $f(.)$ is convex in $\boldsymbol{\Theta}$, it is not possible to prove a convergence theorem to the global minimum. For more specialized cases (e.g., matrix sensing problems under some technical assumptions called restricted isometry property), convergence to a global solution can be characterized with careful initialization procedures (Park et al., 2018; Jain et al., 2013; Zheng & Lafferty, 2016; Park et al., 2016). Since our focus is primarily on neural network applications, where objectives are already nonconvex in $\boldsymbol{\Theta}$, we derive convergence guarantees to a stationary point.

We begin our presentation of the main convergence guarantee by first listing our assumptions. We assume $f_i(\mathbf{U} \mathbf{v}_i)$ are directionally smooth, and that we have access to unbiased stochastic gradients $\tilde{\nabla}_{\mathbf{V}} F(\mathbf{U} \mathbf{V}^\top)$ and $\tilde{\nabla}_{\mathbf{U}} F(\mathbf{U} \mathbf{V}^\top)$ with bounded variance.

**Assumption 1** (Directional smoothness)**.** *We assume that $F(\mathbf{UV}^\top)$ is smooth with respect to $\mathbf{U}$ and $\mathbf{V}$, i.e., there exist constants $L_U, L_V \geq 0$ such that for all $\mathbf{U}_1, \mathbf{U}_2 \in \mathbb{R}^{d \times r}$ and $\mathbf{V}_1, \mathbf{V}_2 \in \mathbb{R}^{n \times r}$:*

$$\left\| \nabla_{\mathbf{U}} F(\mathbf{U}_1 \mathbf{V}_1^\top) - \nabla_{\mathbf{U}} F(\mathbf{U}_2 \mathbf{V}_2^\top) \right\|_F \leq L_U \left( \|\mathbf{U}_1 - \mathbf{U}_2\|_F + \|\mathbf{V}_1 - \mathbf{V}_2\|_F \right)$$

$$\left\| \nabla_{\mathbf{V}} F(\mathbf{U}_1 \mathbf{V}_1^\top) - \nabla_{\mathbf{V}} F(\mathbf{U}_2 \mathbf{V}_2^\top) \right\|_F \leq L_V \left( \|\mathbf{U}_1 - \mathbf{U}_2\|_F + \|\mathbf{V}_1 - \mathbf{V}_2\|_F \right)$$

**Assumption 2** (Stochastic gradients)**.** *We assume access to an unbiased stochastic gradient estimator with bounded variance, i.e.,, there exists $\sigma < +\infty$ such that for all $\mathbf{U} \in \mathbb{R}^{d \times r}$ and $\mathbf{V} \in \mathbb{R}^{n \times r}$:*

$$\mathbb{E}\left[ \tilde{\nabla} F(\mathbf{UV}^\top) \right] = \nabla F(\mathbf{UV}^\top) \qquad and \qquad \begin{aligned} \mathbb{E}\left[ \left\| \tilde{\nabla}_{\mathbf{U}} F(\mathbf{UV}^\top) - \nabla_{\mathbf{U}} F(\mathbf{UV}^\top) \right\|^2 \right] &\leq \sigma^2 \\ \mathbb{E}\left[ \left\| \tilde{\nabla}_{\mathbf{V}} F(\mathbf{UV}^\top) - \nabla_{\mathbf{V}} F(\mathbf{UV}^\top) \right\|^2 \right] &\leq \sigma^2. \end{aligned}$$

**Theorem 1.** *Consider problem* (6) *with smooth loss functions $f_i(.)$ in the sense that Assumption 1 holds. Assume access to a stochastic gradient estimator such that Assumption 2 holds. Furthermore, assume that every client participates in each round with probability $p$ and performs $K$ local steps per iteration. Then, the sequence $\mathbf{U}^t, \mathbf{V}^t$ generated by pFL$^{MF}$ with step-sizes $\eta_v = \frac{p\eta_u}{K}$ and $\eta_u < \frac{1}{2L}$, where $L := \max\{L_U, L_V\}$, satisfies the following bound:*

$$\frac{1}{T}\sum_{t=0}^{T-1} \left( \mathbb{E}\left[ \left\| \nabla_{\mathbf{U}} F(\mathbf{U}^t \mathbf{V}^{t\top}) \right\|^2 \right] + \mathbb{E}\left[ \frac{1}{K}\sum_{k=0}^{K-1} \left\| \nabla_{\mathbf{V}} F(\mathbf{U}^t \mathbf{V}_k^{t\top}) \right\|^2 \right] \right) \leq \frac{2\left( F(\mathbf{U}^0 \mathbf{V}^{0\top}) - F^\star \right)}{\eta T \left( 1 - 2\eta L \right)} + \frac{2\eta L \sigma^2}{1 - 2\eta L}.$$

**Corollary 2.** *Choosing $\eta = \frac{1}{2L\sqrt{T}}$ in Theorem 1 yields a rate of $\mathcal{O}(1/\sqrt{T})$ in the stochastic setting. If full gradients are available ($\sigma = 0$), then $\eta = \frac{1}{4L}$ results in a convergence rate of $\mathcal{O}(1/T)$.*

## 5 Numerical Experiments

In this section, we evaluate the performance of pFL$^{MF}$. We compare the performance of pFL$^{MF}$ against several baselines, including LOCAL training, FEDAVG (McMahan et al., 2017), FEDPER (Arivazhagan et al., 2019), FEDREP (Collins et al., 2021), APFL (Deng et al., 2020), CFL (Sattler et al., 2021), FLUTE (Liu et al., 2024b), FEDAS (Yang et al., 2024), FEDALT (Pillutla et al., 2022), and pFEDFDA (McLaughlin & Su, 2024) by implementing pFL$^{MF}$ in the *FL-Bench* benchmark (Tan & Wang, 2024; Tan et al., 2023). The source code is available at `https://github.com/DadrasAli/pFLMF`. We conducted experiments in five different setups:

**Setup (1)** For the MNIST, CIFAR10, and CIFAR100 datasets, we split the data according to the Dirichlet distribution $\text{Dir}(0.5)$ and $\text{Dir}(1)$ across 100 clients. The labels' distribution is shown in Figures 2c and 2d. Performance of the algorithms is shown in Table 1.

**Setup (2)** For the CIFAR-100 dataset, we partitioned the 100 classes into 20 groups, each containing 5 distinct labels. Data was then distributed among 500 clients, with each client exclusively assigned data from a single group, resulting in highly heterogeneous data. Results are shown in Table 2.

**Setup (3)** For the MNIST, we follow the experimental setup in (Sattler et al., 2019) and consider 1000 clients divided into 10 groups, and labels in each group are re-mapped (permuted) according to a random permutation map. In other words, clients in group one would have the same numbers $\{0, \cdots, 9\}$ but labeled differently; group one may consider 0 with label 0, and group two may consider 0 with label 8. Figures 2a and 2b show the distribution of the labels before and after re-labeling, respectively. Results are shown in Table 2.

**Setup (4)** We sampled a subset of clients, 30% of the total clients, from FEMNIST dataset without changing the underlying data distribution, then we removed clients with less than 10 data points. The remaining set has 1091 clients. We ran the experiments for 1 and 5 numbers of local epochs. Results are shown in Table 2.

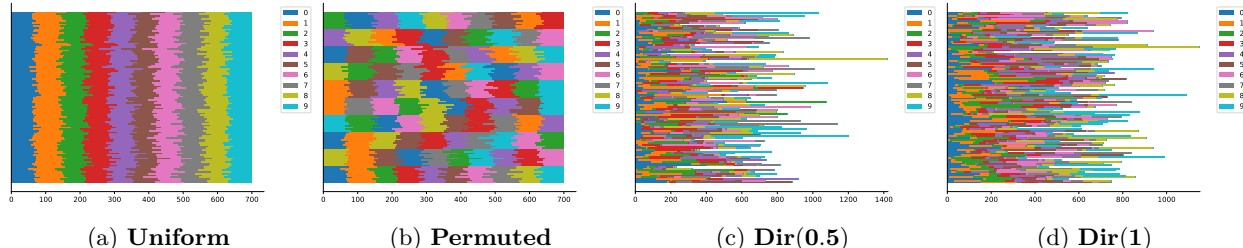

|  | (a) **Uniform** | (b) **Permuted** | (c) **Dir(0.5)** | (d) **Dir(1)** |

Figure 2: Distribution of the labels for MNIST dataset and 100 clients. The vertical and horizontal axes show clients and the size of each client's data, respectively.

| | MNIST | | CIFAR10 | | CIFAR100 | |
|---|---|---|---|---|---|---|
| | **Dir(0.5)** | **Dir(1)** | **Dir(0.5)** | **Dir(1)** | **Dir(0.5)** | **Dir(1)** |
| **Local** | 92.12% ($\pm$0.59) | 89.15% ($\pm$1.12) | 59.14% ($\pm$3.74) | 48.53% ($\pm$2.53) | 16.09% ($\pm$1.52) | 10.66% ($\pm$0.98) |
| **FedAvg** | 96.92% ($\pm$0.65) | 97.01% ($\pm$0.54) | 65.21% ($\pm$2.11) | 65.44% ($\pm$1.68) | 28.30% ($\pm$1.82) | 28.36% ($\pm$1.32) |
| **FedPer** | 96.30% ($\pm$0.26) | 95.16% ($\pm$0.58) | 66.86% ($\pm$3.19) | 58.25% ($\pm$2.22) | 19.98% ($\pm$1.6) | 14.22% ($\pm$1.01) |
| **FedRep** | 95.04% ($\pm$0.40) | 93.33% ($\pm$0.94) | 65.16% ($\pm$3.44) | 55.4% ($\pm$2.06) | 17.49% ($\pm$1.12) | 12.14% ($\pm$1.05) |
| **APFL** | 97.93% ($\pm$0.51) | 97.64% ($\pm$0.39) | 65.99% ($\pm$2.06) | 65.14% ($\pm$1.54) | 27.07% ($\pm$1.57) | 27.07% ($\pm$1.36) |
| **CFL** | 96.92% ($\pm$0.72) | 97.04% ($\pm$0.5) | 64.97% ($\pm$2.68) | 65.98% ($\pm$1.70) | 27.02% ($\pm$1.48) | 24.84% ($\pm$0.91) |
| **FLUTE** | 76.72% ($\pm$0.88) | 73.25% ($\pm$1.01) | 48.20% ($\pm$0.74) | 40.59% ($\pm$0.69) | 12.58% ($\pm$0.28) | 7.28% ($\pm$0.13) |
| **FedAlt** | 98.09% ($\pm$0.05) | 97.91% ($\pm$0.05) | 62.89% ($\pm$0.57) | 58.09% ($\pm$0.43) | 20.57% ($\pm$0.46) | 16.20% ($\pm$0.19) |
| **FedAS** | 97.17% ($\pm$0.12) | 97.11% ($\pm$0.39) | 66.33% ($\pm$1.14) | 63.80% ($\pm$0.54) | 8.48% ($\pm$1.79) | 5.12% ($\pm$1.56) |
| **pFedFDA** | 97.23% ($\pm$0.04) | 97.05% ($\pm$0.08) | **70.65**% ($\pm$1,59) | 66.67% ($\pm$1.41) | 26.15% ($\pm$0.19) | 19.10% ($\pm$0.17) |
| **pFL^MF** | | | | | | |
| $r = 1$ | 96.75% ($\pm$0.61) | 96.53% ($\pm$0.59) | 43.89% ($\pm$3.49) | 64.03% ($\pm$1.66) | 34.32% ($\pm$1.96) | 35.24% ($\pm$1.77) |
| $r = 5$ | 96.78% ($\pm$0.51) | 96.55% ($\pm$0.60) | 60.73% ($\pm$2.86) | 65.89% ($\pm$1.88) | 35.64% ($\pm$2.09) | 35.75% ($\pm$1.24) |
| $r = 10$ | 96.98% ($\pm$0.70) | 96.84% ($\pm$0.56) | 65.10% ($\pm$2.30) | **67.68**% ($\pm$1.56) | 35.28% ($\pm$1.73) | **36.84**% ($\pm$1.50) |
| $r = 15$ | **98.24**% ($\pm$0.26) | **97.93**% ($\pm$0.22) | 68.13% ($\pm$2.43) | 65.88% ($\pm$1.62) | **35.70**% ($\pm$1.77) | 36.12% ($\pm$1.46) |

Table 1: Performance of the algorithms for **Setup (1)**. The best accuracy is shown in boldface, and the second best is underlined.

**Setup (5)** We examine the sensitivity of pFL^MF to the choice of rank, ranging from 1 to 20, on a highly heterogeneous data split, Dir(0.1) and 100 clients. Figure 3 shows the average test accuracy and runtime versus rank for the MNIST and CIFAR-10 datasets.

**Setup (6)** We evaluated pFL^MF with a range of local update counts, $K_U \in \{1, 10, 20\}$ and $r = \{1, 10, 20\}$, on a highly heterogeneous MNIST split generated by Dir(0.1). The experiment involved 200 clients, each communication round sampling a 10% subset. Figure 4 reports the resulting average test accuracy.

**Model.** We used a three-layer neural network, consisting of three linear layers, on the MNIST and FEMNIST datasets and a four-layer convolutional neural network, consisting of two convolutional layers followed by two linear layers, on the CIFAR10 and CIFAR100 datasets. For FedPer and FedRep, we treated the last layer as the classifier, while in pFL^MF, we factorized the entire model.

**Hyper-parameters.** We consider partial participation with probability equal to 0.1. We set the batch size equal to 256 for all algorithms. We tried parameter $r$ values of (6) is set to $r \in \{1, 5, 10, 15\}$. All experiments have 75% train and 25% test data splits on each client's data. We chose the best step size for each algorithm from the set $\{10^{-4}, 10^{-3}, 10^{-2}, 10^{-1}\}$.

**Observations.** In the heterogeneous experiments, Setup (1), pFL^MF outperforms other pFL methods in most cases, though the performance across algorithms is comparable on the MNIST dataset. Remarkably, pFL^MF achieves substantial improvements in average test accuracy when different client groups have similar intra-group distributions but differ significantly across groups (see Table 2). It is important to note that the low test accuracy observed in the CIFAR-100 experiment, Setup (2), is attributed to the simplicity of the neural network model rather than the algorithms themselves. Additionally, the results indicate that multiple

| | MNIST (permuted labels) | CIFAR100 (super groups) | FEMNIST | |
| --- | --- | --- | --- | --- |
| | 1000 clients | 500 clients | 1091 clients | |
| | 1 local epoch | 1 local epoch | 1 local epoch | 5 local epochs |
| **Local** | 25.36% (±0.013) | 10.49% (±0.95) | 50.77% (±0.053) | 65.69%(0.012) |
| **FedAvg** | 12.02% (±0.022) | 36.40% (±1.31) | 65.40% (±0.017) | **77.19**%(0.013) |
| **FedPer** | 19.86% (±0.141) | 14.80% (±0.81) | 66.05% (±0.010) | 67.72%(0.009) |
| **FedRep** | 21.30% (±0.148) | 12.18% (±0.80) | 66.10% (±0.013) | 66.29%(±0.010) |
| **pFL$^{MF}$** | | | | |
| $r = 1$ | 14.70% (±0.083) | 42.94% (±1.22) | 67.82% (±0.134) | 71.42%(0.044) |
| $r = 5$ | 23.75% (±0.027) | 44.70% (±1.91) | 69.99% (±0.123) | 72.09%(±0.208) |
| $r = 10$ | 34.23% (±0.090) | **45.57**% (±1.97) | 72.56% (±0.023) | 72.47%(±0.010) |
| $r = 15$ | **39.31**% (±0.042) | 45.43% (±1.23) | **73.59**% (0.092) | 76.41%(±0.006) |

Table 2: Performance of the algorithms for **Setup (2)**, **Setup (3)**, and **Setup (4)**. The best accuracy is shown in boldface, and the second best is underlined.

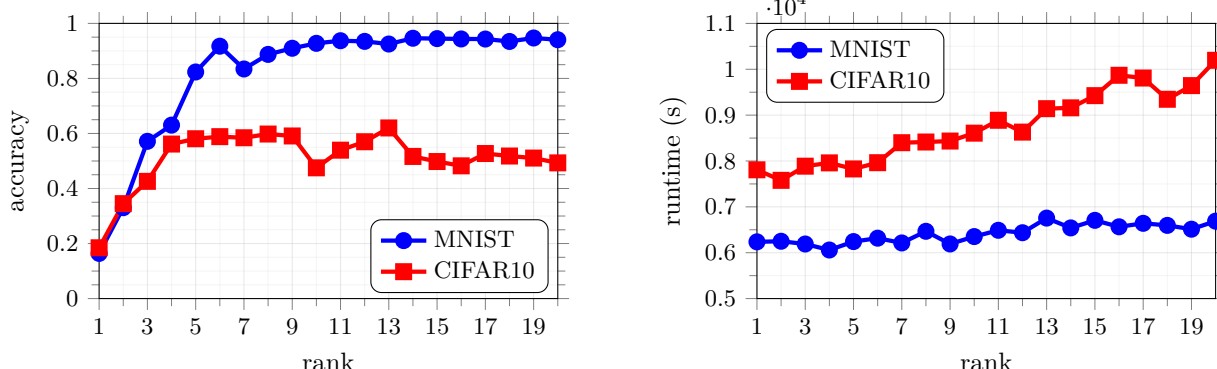

Figure 3: Sensitivity analysis of pFL$^{MF}$ with respect to the factorization rank $r$. The left plot shows prediction accuracy versus rank, while the right plot shows runtime versus rank. Results are averaged over 5 random trials. See **Setup (5)** for details.

local updates in pFL$^{MF}$ contribute positively to performance. Overall, the findings highlight the practical significance of the proposed method.

## 6 Conclusions

We introduced a new pFL formulation based on low-rank matrix optimization and developed a novel pFL algorithm utilizing Burer-Monteiro factorization. We further established convergence guarantees for the proposed method: for minimizing a smooth non-convex objective, the algorithm converges to a stationary point at a rate of $\mathcal{O}(1/T)$ with full gradients; and $\mathcal{O}(1/\sqrt{T})$ for the stochastic setting. Evaluations across four experimental setups highlight the practical significance of the proposed method, especially in scenarios where personalization is essential, and standard approaches are unable to adequately capture the complexity of the underlying data distributions.

We conclude by listing some limitations and future directions. Our numerical experiments demonstrate improved performance of pFL$^{MF}$ with multiple local steps; however, this enhancement is not reflected in our theoretical convergence guarantees. Establishing stronger guarantees that reflect this behavior is a valuable direction for future research. Another notable limitation is that our formulation currently factorizes the entire model (decision variable), which can be computationally intensive in some cases, particularly in large-scale neural network applications. A more efficient approach might be to apply the BM factorization selectively, targeting only a subset of the parameters, which could reduce overhead while maintaining its benefits. Exploring such partial factorizations is a promising direction for future research

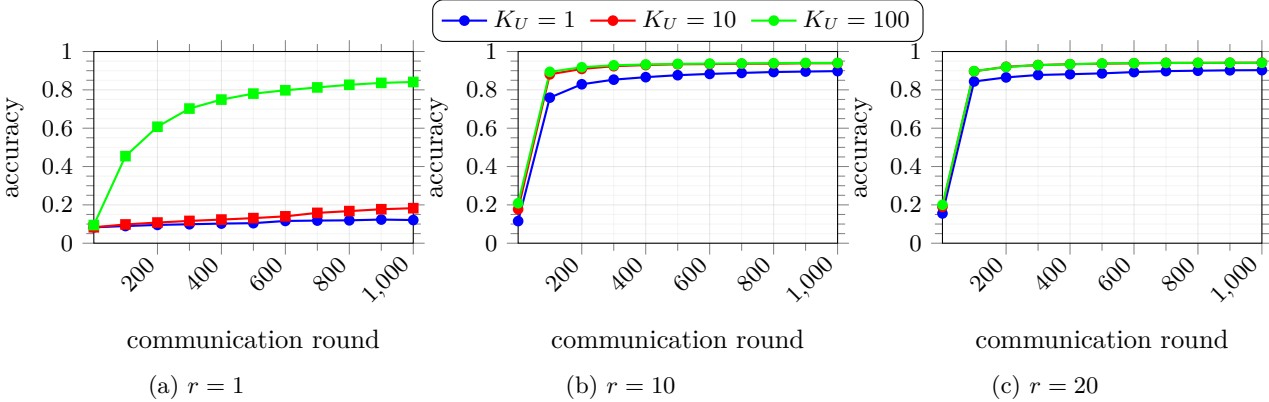

Figure 4: Performance of pFL^MF for different numbers of **U** steps and ranks. See **Setup (6)** for details.

## Acknowledgments

This work was partially conducted while Ali Dadras was at a research visit at CISPA; we sincerely appreciate their support and the stimulating research environment they provided. Alp Yurtsever and Ali Dadras were supported by the Wallenberg AI, Autonomous Systems and Software Program (WASP) funded by the Knut and Alice Wallenberg Foundation. The computations were enabled by the supercomputing resource Berzelius provided by National Supercomputer Centre at Linköping University and the Knut and Alice Wallenberg foundation. Alp Yurtsever and Ali Dadras further acknowledge support from the Swedish Research Council, under registration number 2023-05476.

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

# A  Theoretical Results

## A.1  Compact Notation

We assume that each client participates in the learning process independently with probability $p$. To model this, we define a partial participation matrix $D_t$ for each time step $t$ as a diagonal matrix, where each diagonal entry $[D_t]_{i,i}$ represents the participation status of client $i$ at time $t$. Specifically,

$$[D_t]_{i,i} := \begin{cases} 1, & \text{with probability } p, \\ 0, & \text{with probability } 1-p, \end{cases}$$

where each $[D_t]_{i,i}$ is an independent Bernoulli random variable with parameter $p$. This implies that for each client $i$, $[D_t]_{i,i} = 1$ if the client participates in the training process at time $t$, and $[D_t]_{i,i} = 0$ otherwise. We can write our algorithm in the compact form as follows:

$$\mathbf{V}^{t,0} = \mathbf{V}^t$$
$$\text{for } k = 0, \ldots, K-1, \text{ do}$$
$$\mathbf{V}^t_{k+1} = \mathbf{V}^t_k - \eta_v \mathbf{D}_t \tilde{\nabla}_{\mathbf{V}} F(\mathbf{U}^t \mathbf{V}^t_k{}^\top)$$
$$\text{end for}$$
$$\mathbf{U}^{t+1} = \mathbf{U}^t - \eta_u \tilde{\nabla}_{\mathbf{U}} F(\mathbf{U}^t \mathbf{V}^t{}^\top)$$
$$\mathbf{V}^{t+1} = \mathbf{V}^t_K .$$

We define expectations with respect to gradient noise as $\mathbb{E}^{\mathbf{V}}_{\text{noise}}[\cdot]$ and $\mathbb{E}^{\mathbf{U}}_{\text{noise}}[\cdot]$, and expectation with respect to participation randomness as $\mathbb{E}_{\mathcal{S}_t}[\cdot]$. For participation randomness, we assume that $\mathbb{E}_{\mathcal{S}_t}[\mathbf{D}_t] = p\mathbf{I}$, where $\mathbf{I}$ is the identity matrix and $p$ is the probability of client participation under independent sampling. We define the conditional expectation given all randomness before iteration $t$ and local step $k$ as

$$\mathbb{E}_{t,k}[\cdot] := \mathbb{E}^{\mathbf{V}}_{\text{noise}}\Big[ \cdot \mid \text{randomness before } (t,k), \mathbf{D}_t \Big],$$

where the randomness includes all prior gradient noise and participation randomness up to local step $k$ of iteration $t$. Additionally, we define the conditional expectation given all randomness in the algorithm before iteration $t$ and the final local step $K$ as

$$\mathbb{E}_t[\cdot] := \mathbb{E}^{\mathbf{U}}_{\text{noise}}\Big[ \cdot \mid \text{randomness before } (t,K), \mathbf{D}_t \Big].$$

Finally, we use $\mathbb{E}[\cdot]$ to denote the total expectation over all sources of randomness in the algorithm, including gradient noise and client participation.

## A.2  Convergence Analysis

We start with proving some useful bounds. For any $t$,

$$\mathbb{E}_t\left[ \left\| \tilde{\nabla}_{\mathbf{U}} F(\mathbf{U}^t \mathbf{V}^t{}^\top) \right\|^2_F \right] = \mathbb{E}_t\left[ \left\| \nabla_{\mathbf{U}} F(\mathbf{U}^t \mathbf{V}^t{}^\top) \right\|^2_F + \left\| \tilde{\nabla}_{\mathbf{U}} F(\mathbf{U}^t \mathbf{V}^t{}^\top) - \nabla_{\mathbf{U}} F(\mathbf{U}^t \mathbf{V}^t{}^\top) \right\|^2_F \right.$$
$$\left. + 2 \left\langle \nabla_{\mathbf{U}} F(\mathbf{U}^t \mathbf{V}^t{}^\top), \tilde{\nabla}_{\mathbf{U}} F(\mathbf{U}^t \mathbf{V}^t{}^\top) - \nabla_{\mathbf{U}} F(\mathbf{U}^t \mathbf{V}^t{}^\top) \right\rangle \right]$$
$$\leq \left\| \nabla_{\mathbf{U}} F(\mathbf{U}^t \mathbf{V}^t{}^\top) \right\|^2_F + \sigma^2. \tag{10}$$

Similar to above, we can write, for any $t$ and $k$,

$$
\begin{aligned}
\mathbb{E}_{t,k}\left[\left\|\mathbf{D}_t\tilde{\nabla}_{\mathbf{V}}F(\mathbf{U}^t\mathbf{V}_k^{t\top})\right\|_F^2\right] &= \mathbb{E}_{t,k}\left[\left\|\mathbf{D}_t\nabla_{\mathbf{V}}F(\mathbf{U}^t\mathbf{V}_k^{t\top})\right\|_F^2 + \left\|\mathbf{D}_t\tilde{\nabla}_{\mathbf{V}}F(\mathbf{U}^t\mathbf{V}_k^{t\top}) - \mathbf{D}_t\nabla_{\mathbf{V}}F(\mathbf{U}^t\mathbf{V}_k^{t\top})\right\|_F^2 \right.\\
&\qquad\left. + 2\left\langle \mathbf{D}_t\nabla_{\mathbf{V}}F(\mathbf{U}^t\mathbf{V}_k^{t\top}), \underbrace{\mathbf{D}_t\tilde{\nabla}_{\mathbf{V}}F(\mathbf{U}^t\mathbf{V}_k^{t\top}) - \mathbf{D}_t\nabla_{\mathbf{V}}F(\mathbf{U}^t\mathbf{V}_k^{t\top})}_{\mathbb{E}_{t,k}[\ .\ |\mathbf{D}_t]=0}\right\rangle\right]\\
&= \left\|\mathbf{D}_t\nabla_{\mathbf{V}}F(\mathbf{U}^t\mathbf{V}_k^{t\top})\right\|_F^2 + \|\mathbf{D}_t\|_2^2\cdot\mathbb{E}_{t,k}\left[\left\|\tilde{\nabla}_{\mathbf{V}}F(\mathbf{U}^t\mathbf{V}_k^{t\top}) - \nabla_{\mathbf{V}}F(\mathbf{U}^t\mathbf{V}_k^{t\top})\right\|_F^2\right]\\
&= \left\|\mathbf{D}_t\nabla_{\mathbf{V}}F(\mathbf{U}^t\mathbf{V}_k^{t\top})\right\|_F^2 + \|\mathbf{D}_t\|_2^2\cdot\sigma^2
\end{aligned}
$$

where in the third line, we used the submultiplicative property of the Frobenius norm. Now we take the expectation with respect to the participation probability

$$
\begin{aligned}
\mathbb{E}_{\mathcal{S}_t}\left[\mathbb{E}_{t,k}\left[\left\|\mathbf{D}_t\tilde{\nabla}_{\mathbf{V}}F(\mathbf{U}^t\mathbf{V}_k^{t\top})\right\|_F^2\right]\right] &= \mathbb{E}_{\mathcal{S}_t}\left[\left\|\mathbf{D}_t\nabla_{\mathbf{V}}F(\mathbf{U}^t\mathbf{V}_k^{t\top})\right\|_F^2 + \|\mathbf{D}_t\|_2^2\cdot\sigma^2\right]\\
&\leq p\left\|\nabla_{\mathbf{V}}F(\mathbf{U}^t\mathbf{V}^{t\top})\right\|_F^2 + \sigma^2\,,
\end{aligned}
\tag{11}
$$

where we used Lemma 6, and Lemma 3.

**(A)** First, we use the smoothness of $F(\mathbf{U}\mathbf{V}^{t\top})$ with respect to $\mathbf{V}$ and write

$$
\begin{aligned}
F(\mathbf{U}^t\mathbf{V}_{k+1}^{t\top}) &\leq F(\mathbf{U}^t\mathbf{V}_k^{t\top}) + \left\langle\nabla_{\mathbf{V}}F(\mathbf{U}^t\mathbf{V}_k^{t\top}), \mathbf{V}_{k+1}^t - \mathbf{V}_k^t\right\rangle + \frac{L_V}{2}\left\|\mathbf{V}_{k+1}^t - \mathbf{V}_k^t\right\|_F^2\\
&= F(\mathbf{U}^t\mathbf{V}_k^{t\top}) - \eta_v\left\langle\nabla_{\mathbf{V}}F(\mathbf{U}^t\mathbf{V}_k^{t\top}), \mathbf{D}_t\tilde{\nabla}_{\mathbf{V}}F(\mathbf{U}^t\mathbf{V}_k^{t\top})\right\rangle + \eta_v^2\frac{L_V}{2}\left\|\mathbf{D}_t\tilde{\nabla}_{\mathbf{V}}F(\mathbf{U}^t\mathbf{V}_k^{t\top})\right\|_F^2.
\end{aligned}
$$

Taking conditional expectation, we get

$$
\begin{aligned}
\mathbb{E}_{\mathcal{S}_t}\left[\mathbb{E}_{t,k}\left[F(\mathbf{U}^t\mathbf{V}_{k+1}^{t\top})\right]\right] &\leq F(\mathbf{U}^t\mathbf{V}_k^{t\top}) - \eta_v p\left\|\nabla_{\mathbf{V}}F(\mathbf{U}^t\mathbf{V}_k^{t\top})\right\|_F^2 + \eta_v^2\frac{pL_V}{2}\left\|\nabla_{\mathbf{V}}F(\mathbf{U}^t\mathbf{V}_k^{t\top})\right\|_F^2 + \eta_v^2\frac{L_V}{2}\sigma^2\\
&= F(\mathbf{U}^t\mathbf{V}_k^{t\top}) - \eta_v p\left(1 - \eta_v\frac{L_V}{2}\right)\left\|\nabla_{\mathbf{V}}F(\mathbf{U}^t\mathbf{V}_k^{t\top})\right\|_F^2 + \eta_v^2\frac{L_V\sigma^2}{2}
\end{aligned}
$$

where we used (11) in the second line. We rearrange the inequality above and average over $k$ to obtain

$$
\begin{aligned}
\eta_v p\left(1 - \eta_v\frac{L_V}{2}\right)&\frac{1}{K}\sum_{k=0}^{K-1}\left\|\nabla_{\mathbf{V}}F(\mathbf{U}^t\mathbf{V}_k^{t\top})\right\|_F^2\\
&\leq \frac{1}{K}\left(F(\mathbf{U}^t\mathbf{V}^{t,0\top}) - \mathbb{E}_{\mathcal{S}_t}\left[\mathbb{E}_{t,K}\left[F(\mathbf{U}^t\mathbf{V}^{t,K\top})\right]\right]\right) + \eta_v^2\frac{L_V\sigma^2}{2}\\
&= \frac{1}{K}\left(F(\mathbf{U}^t\mathbf{V}^{t\top}) - \mathbb{E}_{\mathcal{S}_t}\left[\mathbb{E}_{t,K}\left[F(\mathbf{U}^t\mathbf{V}^{t+1\top})\right]\right]\right) + \eta_v^2\frac{L_V\sigma^2}{2}
\end{aligned}
\tag{12}
$$

where, in the second line, we used $\mathbf{V}^{t,0} = \mathbf{V}^t$ and $\mathbf{V}^{t,K} = \mathbf{V}^{t+1}$.

**(B)** Now, we will use the smoothness again, but this time with respect to $\mathbf{U}$:

$$
\begin{aligned}
F(\mathbf{U}^{t+1}\mathbf{V}_k^{t\top}) &\leq F(\mathbf{U}^t\mathbf{V}_k^{t\top}) + \left\langle\nabla_{\mathbf{U}}F(\mathbf{U}^t\mathbf{V}_k^{t\top}), \mathbf{U}^{t+1} - \mathbf{U}^t\right\rangle + \frac{L_U}{2}\left\|\mathbf{U}^{t+1} - \mathbf{U}^t\right\|_F^2\\
&\leq F(\mathbf{U}^t\mathbf{V}_k^{t\top}) - \eta_u\left\langle\nabla_{\mathbf{U}}F(\mathbf{U}^t\mathbf{V}_k^{t\top}), \tilde{\nabla}_{\mathbf{U}}F(\mathbf{U}^t\mathbf{V}^{t\top})\right\rangle + \eta_u^2\frac{L_U}{2}\left\|\tilde{\nabla}_{\mathbf{U}}F(\mathbf{U}^t\mathbf{V}^{t\top})\right\|_F^2
\end{aligned}
$$

Similar to the previous case, if we take expectation with respect to the randomness in $\mathbf{U}$ update at iteration $t$, we obtain the following bound by using (10):

$$
\mathbb{E}_t\left[F(\mathbf{U}^{t+1}\mathbf{V}_k^{t\top})\right] \leq F(\mathbf{U}^t\mathbf{V}_k^{t\top}) - \eta_u\left\langle\nabla_{\mathbf{U}}F(\mathbf{U}^t\mathbf{V}_k^{t\top}), \nabla_{\mathbf{U}}F(\mathbf{U}^t\mathbf{V}^{t\top})\right\rangle + \eta_u^2\frac{L_U}{2}\left\|\nabla_{\mathbf{U}}F(\mathbf{U}^t\mathbf{V}^{t\top})\right\|_F^2 + \eta_u^2\frac{L_U\sigma^2}{2}\,.
$$

If we split the inner product term as

$$\left\langle \nabla_{\mathbf{U}} F(\mathbf{U}^t \mathbf{V}_k^{t\top}), \nabla_{\mathbf{U}} F(\mathbf{U}^t \mathbf{V}^{t\top}) \right\rangle = \left\langle \nabla_{\mathbf{U}} F(\mathbf{U}^t \mathbf{V}_k^{t\top}), \nabla_{\mathbf{U}} F(\mathbf{U}^t \mathbf{V}^{t\top}) - \nabla_{\mathbf{U}} F(\mathbf{U}^t \mathbf{V}_k^{t\top}) + \nabla_{\mathbf{U}} F(\mathbf{U}^t \mathbf{V}_k^{t\top}) \right\rangle$$

$$= \left\langle \nabla_{\mathbf{U}} F(\mathbf{U}^t \mathbf{V}_k^{t\top}), \nabla_{\mathbf{U}} F(\mathbf{U}^t \mathbf{V}^{t\top}) - \nabla_{\mathbf{U}} F(\mathbf{U}^t \mathbf{V}_k^{t\top}) \right\rangle + \left\| \nabla_{\mathbf{U}} F(\mathbf{U}^t \mathbf{V}_k^{t\top}) \right\|_F^2$$

$$\geq -\frac{\eta_u L_U}{2} \left\| \nabla_{\mathbf{U}} F(\mathbf{U}^t \mathbf{V}_k^{t\top}) \right\|_F^2 - \frac{1}{2\eta_u L_U} \left\| \nabla_{\mathbf{U}} F(\mathbf{U}^t \mathbf{V}^{t\top}) - \nabla_{\mathbf{U}} F(\mathbf{U}^t \mathbf{V}_k^{t\top}) \right\|_F^2 + \left\| \nabla_{\mathbf{U}} F(\mathbf{U}^t \mathbf{V}_k^{t\top}) \right\|_F^2,$$

where the last line follows from Young's inequality (17) with $\alpha = \eta_u L_U$. Moreover, by the smoothness assumption, we have

$$\frac{1}{2L_U \eta_u} \left\| \nabla_{\mathbf{U}} F(\mathbf{U}^t \mathbf{V}^{t\top}) - \nabla_{\mathbf{U}} F(\mathbf{U}^t \mathbf{V}_k^{t\top}) \right\|^2 \leq \frac{L_U}{2\eta_u} \left\| \mathbf{V}^t - \mathbf{V}_k^t \right\|^2 \leq \frac{\eta_v^2}{\eta_u} \frac{L_U}{2} \left\| \sum_{i=0}^{k-1} \nabla_{\mathbf{V}} F(\mathbf{U}^t \mathbf{V}_i^{t\top}) \right\|_F^2.$$

Combining all these bounds, we get

$$\mathbb{E}_t \left[ F(\mathbf{U}^{t+1} \mathbf{V}_k^{t\top}) \right] \leq F(\mathbf{U}^t \mathbf{V}_k^{t\top}) - \eta_u \left( 1 - \eta_u \frac{L_U}{2} \right) \left\| \nabla_{\mathbf{U}} F(\mathbf{U}^t \mathbf{V}_k^{t\top}) \right\|_F^2 + \eta_u^2 \frac{L_U}{2} \left\| \nabla_{\mathbf{U}} F(\mathbf{U}^t \mathbf{V}^{t\top}) \right\|_F^2$$

$$+ \eta_v^2 \frac{L_U}{2} \left\| \sum_{i=0}^{k-1} \nabla_{\mathbf{V}} F(\mathbf{U}^t \mathbf{V}_i^{t\top}) \right\|_F^2 + \eta_u^2 \frac{L_U \sigma^2}{2}. \quad (13)$$

Now we consider two cases $k = 0$ and $k = K$ in (13).

1. For $\mathbf{k = 0}$, we have

$$\mathbb{E}_t \left[ F(\mathbf{U}^{t+1} \mathbf{V}^{t\top}) \right] \leq F(\mathbf{U}^t \mathbf{V}^{t\top}) - \eta_u (1 - \eta_u L_U) \left\| \nabla_{\mathbf{U}} F(\mathbf{U}^t \mathbf{V}^{t\top}) \right\|_F^2 + \eta_u^2 \frac{L_U \sigma^2}{2}. \quad (14)$$

   where we used $\mathbf{V}^{t,0} = \mathbf{V}^t$.

2. For $\mathbf{k = K}$, we have

$$\mathbb{E}_t \left[ F(\mathbf{U}^{t+1} \mathbf{V}^{t+1\top}) \right] \leq F(\mathbf{U}^t \mathbf{V}^{t+1\top}) - \eta_u \left( 1 - \eta_u \frac{L_U}{2} \right) \left\| \nabla_{\mathbf{U}} F(\mathbf{U}^t \mathbf{V}^{t+1\top}) \right\|_F^2 + \eta_u^2 \frac{L_U}{2} \left\| \nabla_{\mathbf{U}} F(\mathbf{U}^t \mathbf{V}^{t\top}) \right\|_F^2$$

$$+ \eta_v^2 \frac{L_U}{2} \left\| \sum_{k=0}^{K-1} \nabla_{\mathbf{V}} F(\mathbf{U}^t \mathbf{V}_k^{t\top}) \right\|_F^2 + \eta_u^2 \frac{L_U \sigma^2}{2}$$

$$\leq F(\mathbf{U}^t \mathbf{V}^{t+1\top}) + \eta_u^2 \frac{L_U}{2} \left\| \nabla_{\mathbf{U}} F(\mathbf{U}^t \mathbf{V}^{t\top}) \right\|_F^2 + \eta_v^2 \frac{L_U}{2} \left\| \sum_{k=0}^{K-1} \nabla_{\mathbf{V}} F(\mathbf{U}^t \mathbf{V}_k^{t\top}) \right\|_F^2 + \eta_u^2 \frac{L_U \sigma^2}{2}.$$

   Rearranging the terms we can write

$$-\frac{L_U}{2} \left( \eta_u^2 \left\| \nabla_{\mathbf{U}} F(\mathbf{U}^t \mathbf{V}^{t\top}) \right\|_F^2 + \eta_v^2 \left\| \sum_{k=0}^{K-1} \nabla_{\mathbf{V}} F(\mathbf{U}^t \mathbf{V}_k^{t\top}) \right\|_F^2 \right)$$

$$\leq F(\mathbf{U}^t \mathbf{V}^{t+1\top}) - \mathbb{E}_t \left[ F(\mathbf{U}^{t+1} \mathbf{V}^{t+1\top}) \right] + \eta_u^2 \frac{L_U \sigma^2}{2}.$$

**(C)** We once again use smoothness with respect to $\mathbf{V}$:

$$F(\mathbf{U}^{t+1} \mathbf{V}_{k+1}^{t\top}) \leq F(\mathbf{U}^{t+1} \mathbf{V}_k^{t\top}) + \left\langle \nabla_{\mathbf{V}} F(\mathbf{U}^{t+1} \mathbf{V}_k^{t\top}), \mathbf{V}_{k+1}^t - \mathbf{V}_k^t \right\rangle + \frac{L_V}{2} \left\| \mathbf{V}_{k+1}^t - \mathbf{V}_k^t \right\|_F^2$$

$$= F(\mathbf{U}^{t+1} \mathbf{V}_k^{t\top}) - \eta_v \left\langle \nabla_{\mathbf{V}} F(\mathbf{U}^{t+1} \mathbf{V}_k^{t\top}), \mathbf{D}_t \tilde{\nabla}_{\mathbf{V}} F(\mathbf{U}^t \mathbf{V}_k^{t\top}) \right\rangle + \eta_v^2 \frac{L_V}{2} \left\| \mathbf{D}_t \tilde{\nabla}_{\mathbf{V}} F(\mathbf{U}^t \mathbf{V}_k^{t\top}) \right\|_F^2.$$

We take the conditional expectation

$$\mathbb{E}_{\mathcal{S}_t}\left[\mathbb{E}_{t,k}\left[F(\mathbf{U}^{t+1}\mathbf{V}_{k+1}^t{}^\top)\right]\right] \le F(\mathbf{U}^{t+1}\mathbf{V}_k^t{}^\top) - \eta_v\mathbb{E}_{\mathcal{S}_t}\left[\mathbb{E}_{t,k}\left[\left\langle\nabla_\mathbf{V}F(\mathbf{U}^{t+1}\mathbf{V}_k^t{}^\top),\mathbf{D}_t\tilde\nabla_\mathbf{V}F(\mathbf{U}^t\mathbf{V}_k^t{}^\top)\right\rangle\right]\right]$$
$$+ \eta_v^2\frac{pL_V}{2}\left\|\nabla_\mathbf{V}F(\mathbf{U}^t\mathbf{V}_k^t{}^\top)\right\|_F^2 + \eta_v^2\frac{L_V\sigma^2}{2},$$

where we used (11). Focusing again on the inner product term, we obtain

$$\mathbb{E}_{\mathcal{S}_t}\left[\mathbb{E}_{t,k}\left[\left\langle\nabla_\mathbf{V}F(\mathbf{U}^{t+1}\mathbf{V}_k^t{}^\top),\mathbf{D}_t\tilde\nabla_\mathbf{V}F(\mathbf{U}^t\mathbf{V}_k^t{}^\top)\right\rangle\right]\right]$$
$$= \mathbb{E}_{\mathcal{S}_t}\left[\mathbb{E}_{t,k}\left[\left\langle\nabla_\mathbf{V}F(\mathbf{U}^{t+1}\mathbf{V}_k^t{}^\top)\pm\nabla_\mathbf{V}F(\mathbf{U}^t\mathbf{V}_k^t{}^\top),\mathbf{D}_t\tilde\nabla_\mathbf{V}F(\mathbf{U}^t\mathbf{V}_k^t{}^\top)\right\rangle\right]\right]$$
$$= \mathbb{E}_{\mathcal{S}_t}\left[\left\langle\nabla_\mathbf{V}F(\mathbf{U}^{t+1}\mathbf{V}_k^t{}^\top)-\nabla_\mathbf{V}F(\mathbf{U}^t\mathbf{V}_k^t{}^\top),\mathbf{D}_t\nabla_\mathbf{V}F(\mathbf{U}^t\mathbf{V}_k^t{}^\top)\right\rangle + \left\langle\nabla_\mathbf{V}F(\mathbf{U}^t\mathbf{V}_k^t{}^\top),\mathbf{D}_t\nabla_\mathbf{V}F(\mathbf{U}^t\mathbf{V}_k^t{}^\top)\right\rangle\right]$$
$$= p\left\langle\nabla_\mathbf{V}F(\mathbf{U}^{t+1}\mathbf{V}_k^t{}^\top)-\nabla_\mathbf{V}F(\mathbf{U}^t\mathbf{V}_k^t{}^\top),\nabla_\mathbf{V}F(\mathbf{U}^t\mathbf{V}_k^t{}^\top)\right\rangle + p\left\|\nabla_\mathbf{V}F(\mathbf{U}^t\mathbf{V}_k^t{}^\top)\right\|_F^2$$
$$\ge -\frac{p}{2\eta_v KL_V}\left\|\nabla_\mathbf{V}F(\mathbf{U}^{t+1}\mathbf{V}_k^t{}^\top)-\nabla_\mathbf{V}F(\mathbf{U}^t\mathbf{V}_k^t{}^\top)\right\|_F^2 - \frac{\eta_v pKL_V}{2}\left\|\nabla_\mathbf{V}F(\mathbf{U}^t\mathbf{V}_k^t{}^\top)\right\|_F^2 + p\left\|\nabla_\mathbf{V}F(\mathbf{U}^t\mathbf{V}_k^t{}^\top)\right\|_F^2$$
$$\ge -\frac{pL_V}{2\eta_v K}\left\|\mathbf{U}^{t+1}-\mathbf{U}^t\right\|_F^2 - \frac{\eta_v pKL_V}{2}\left\|\nabla_\mathbf{V}F(\mathbf{U}^t\mathbf{V}_k^t{}^\top)\right\|_F^2 + p\left\|\nabla_\mathbf{V}F(\mathbf{U}^t\mathbf{V}_k^t{}^\top)\right\|_F^2$$
$$= -\frac{pL_V}{2\eta_v K}\eta_u^2\left\|\nabla_\mathbf{U}F(\mathbf{U}^t\mathbf{V}^t{}^\top)\right\|_F^2 - \frac{\eta_v pKL_V}{2}\left\|\nabla_\mathbf{V}F(\mathbf{U}^t\mathbf{V}_k^t{}^\top)\right\|_F^2 + p\left\|\nabla_\mathbf{V}F(\mathbf{U}^t\mathbf{V}_k^t{}^\top)\right\|_F^2,$$

where we used Young's inequality (17) in the fourth line with $\alpha = \eta_v KL_V$. Substituting back, we get

$$\mathbb{E}_{\mathcal{S}_t}\left[\mathbb{E}_{t,k}\left[F(\mathbf{U}^{t+1}\mathbf{V}_{k+1}^t{}^\top)\right]\right] \le F(\mathbf{U}^{t+1}\mathbf{V}_k^t{}^\top) - \eta_v p\left(1 - \eta_v\frac{(K+1)L_V}{2}\right)\left\|\nabla_\mathbf{V}F(\mathbf{U}^t\mathbf{V}_k^t{}^\top)\right\|_F^2$$
$$+ \eta_u^2\frac{pL_V}{2K}\left\|\nabla_\mathbf{U}F(\mathbf{U}^t\mathbf{V}^t{}^\top)\right\|_F^2 + \eta_v^2\frac{L_V\sigma^2}{2}$$

By summing over $k$ and appropriately rearranging the terms in the inequality, we obtain the following:

$$\eta_v p\left(1 - \eta_v\frac{(K+1)L_V}{2}\right)\sum_{k=0}^{K-1}\left\|\nabla_\mathbf{V}F(\mathbf{U}^t\mathbf{V}_k^t{}^\top)\right\|_F^2 - \eta_u^2\frac{pL_V}{2}\left\|\nabla_\mathbf{U}F(\mathbf{U}^t\mathbf{V}^t{}^\top)\right\|^2$$
$$\le F(\mathbf{U}^{t+1}\mathbf{V}^t{}^\top) - \mathbb{E}_{\mathcal{S}_t}\left[\mathbb{E}_{t,K}\left[F(\mathbf{U}^{t+1}\mathbf{V}^{t+1}{}^\top)\right]\right] + \eta_v^2\frac{KL_V\sigma^2}{2}.$$

where we used $\mathbf{V}^{t,K} = \mathbf{V}^{t+1}$ and $\mathbf{V}^{t,0} = \mathbf{V}^t$. We define

$$\Delta_U := \mathbb{E}_t\left[\left\|\nabla_\mathbf{U}F(\mathbf{U}^t\mathbf{V}^t{}^\top)\right\|_F^2\right]$$
$$\Delta_V := \mathbb{E}_{\mathcal{S}_t}\left[\frac{1}{K}\sum_{k=0}^{K-1}\mathbb{E}_{t,k}\left[\left\|\nabla_\mathbf{V}F(\mathbf{U}^t\mathbf{V}_k^t{}^\top)\right\|_F^2\right]\right]. \tag{15}$$

We summarize the resulting inequalities in part **(A)** to **(C)** as

$$\eta_v p\left(1 - \eta_v\frac{L_V}{2}\right)\Delta_V \le \frac{1}{K}\left(F(\mathbf{U}^t\mathbf{V}^t{}^\top) - \mathbb{E}_{\mathcal{S}_t}\left[\mathbb{E}_{t,K}\left[F(\mathbf{U}^t\mathbf{V}^{t+1}{}^\top)\right]\right]\right) + \eta_v^2\frac{L_V\sigma^2}{2}$$

$$\eta_u\left(1 - \eta_u L_U\right)\Delta_U \le F(\mathbf{U}^t\mathbf{V}^t{}^\top) - \mathbb{E}_t\left[F(\mathbf{U}^{t+1}\mathbf{V}^t{}^\top)\right] + \eta_u^2\frac{L_U\sigma^2}{2}$$

$$-\frac{L_U}{2}\left(\eta_u^2\Delta_U + \eta_v^2\left\|\sum_{k=0}^{K-1}\nabla_\mathbf{V}F(\mathbf{U}^t\mathbf{V}_k^t{}^\top)\right\|^2\right) \le F(\mathbf{U}^t\mathbf{V}^{t+1}{}^\top) - \mathbb{E}_t\left[F(\mathbf{U}^{t+1}\mathbf{V}^{t+1}{}^\top)\right] + \eta_u^2\frac{L_U\sigma^2}{2}$$

$$\eta_v p\left(1 - \eta_v\frac{(K+1)L_V}{2}\right)K\Delta_V - \eta_u^2\frac{pL_V}{2}\Delta_U \le F(\mathbf{U}^{t+1}\mathbf{V}^t{}^\top) - \mathbb{E}_{\mathcal{S}_t}\left[\mathbb{E}_{t,K}\left[F(\mathbf{U}^{t+1}\mathbf{V}^{t+1}{}^\top)\right]\right] + \eta_v^2\frac{KL_V\sigma^2}{2}.$$

where we used the definitions (15). We rewrite four inequalities above using $\eta_v = \frac{p\eta_u}{K} := \frac{p\eta}{K}$ and defining $L := \max\{L_U, L_V\}$ as

$$\eta p^2 \big(1 - \eta \frac{pL}{2K}\big) \Delta_V \leq F(\mathbf{U}^t \mathbf{V}^{t\top}) - \mathbb{E}_{\mathcal{S}_t}\Big[\mathbb{E}_{t,K}\big[F(\mathbf{U}^t \mathbf{V}^{t+1\top})\big]\Big] + \eta^2 \frac{L\sigma^2}{2K}$$

$$\eta\big(1 - \eta L\big)\Delta_U \leq F(\mathbf{U}^t \mathbf{V}^{t\top}) - \mathbb{E}_t\big[F(\mathbf{U}^{t+1} \mathbf{V}^{t\top})\big] + \eta^2 \frac{L\sigma^2}{2}$$

$$-\frac{L}{2}\left(\eta^2 \mathbb{E}_t[\Delta_U] + \frac{\eta^2 p^2}{K^2}\left\|\sum_{k=0}^{K-1} \nabla_{\mathbf{V}} F(\mathbf{U}^t \mathbf{V}_k^{t\top})\right\|^2\right) \leq F(\mathbf{U}^t \mathbf{V}^{t+1\top}) - \mathbb{E}_t\big[F(\mathbf{U}^{t+1} \mathbf{V}^{t+1\top})\big] + \eta^2 \frac{L\sigma^2}{2}$$

$$\frac{\eta p^2}{K}\big(1 - \eta p \frac{(K+1)L}{2K}\big)K\Delta_V - \eta^2 \frac{pL}{2}\Delta_U \leq F(\mathbf{U}^{t+1} \mathbf{V}^{t\top}) - \mathbb{E}_{\mathcal{S}_t}\Big[\mathbb{E}_{t,K}\big[F(\mathbf{U}^{t+1} \mathbf{V}^{t+1\top})\big]\Big] + \eta^2 \frac{L\sigma^2}{2K}.$$

Summing up the inequalities above, we get

$$\eta\big(1 - 2\eta L\big)(\Delta_U + \Delta_V) \leq \eta\big(1 - 2\eta L\big)\Delta_U + \eta p\big(1 - \eta \frac{pL}{2K}\big)\Delta_V$$

$$\leq 2\left(F(\mathbf{U}^t \mathbf{V}^{t\top}) - \mathbb{E}_{\mathcal{S}_t}\Big[\mathbb{E}_{t,K}\big[F(\mathbf{U}^{t+1} \mathbf{V}^{t+1\top})\big]\Big]\right) + \eta^2\big(1 + \frac{1}{K}\big)L\sigma^2$$

$$\leq 2\left(F(\mathbf{U}^t \mathbf{V}^{t\top}) - \mathbb{E}_{\mathcal{S}_t}\Big[\mathbb{E}_{t,K}\big[F(\mathbf{U}^{t+1} \mathbf{V}^{t+1\top})\big]\Big]\right) + 2\eta^2 L\sigma^2, \tag{16}$$

where we used the following inequality

$$\eta^2 \frac{p^2 L}{2K^2}\left\|\sum_{k=0}^{K-1} \nabla_{\mathbf{V}} F(\mathbf{U}^t \mathbf{V}_k^{t\top})\right\|^2 - \frac{\eta p^2}{K}\left(1 - \eta p \frac{(K+1)L}{2K}\right)\sum_{k=0}^{K-1}\left\|\nabla_{\mathbf{V}} F(\mathbf{U}^t \mathbf{V}_k^{t\top})\right\|^2$$

$$\leq \eta^2 \frac{p^2 L}{2K^2}K\sum_{k=0}^{K-1}\left\|\nabla_{\mathbf{V}} F(\mathbf{U}^t \mathbf{V}_k^{t\top})\right\|^2 - \frac{\eta p^2}{K}\left(1 - \eta p \frac{(K+1)L}{2K}\right)\sum_{k=0}^{K-1}\left\|\nabla_{\mathbf{V}} F(\mathbf{U}^t \mathbf{V}_k^{t\top})\right\|^2$$

$$\leq \frac{\eta p^2}{K}\left(\eta \frac{L}{2}\big(\frac{(1+p)K+1}{K}\big) - 1\right)\sum_{k=0}^{K-1}\left\|\nabla_{\mathbf{V}} F(\mathbf{U}^t \mathbf{V}_k^{t\top})\right\|^2$$

$$\leq 0 \quad (\eta \leq \frac{1}{2L}),$$

where in the second line, we used Jensen's inequality, see (18). Next, we take the expectation over all sources of randomness in the algorithm. Then, we average both sides of (16) over the iterations $t$, followed by dividing both sides by $\eta(1 - 2\eta L)$, yielding the following expression:

$$\frac{1}{T}\sum_{t=0}^{T-1}\left(\mathbb{E}\left[\left\|\nabla_{\mathbf{U}} F(\mathbf{U}^t \mathbf{V}^{t\top})\right\|^2\right] + \mathbb{E}\left[\frac{1}{K}\sum_{k=0}^{K-1}\left\|\nabla_{\mathbf{V}} F(\mathbf{U}^t \mathbf{V}_k^{t\top})\right\|^2\right]\right)$$

$$\leq \frac{2\left(F(\mathbf{U}^0 \mathbf{V}^{0\top}) - F(\mathbf{U}^T \mathbf{V}^{T\top})\right)}{\eta T\big(1 - 2\eta L\big)} + \frac{2\eta L\sigma^2}{1 - 2\eta L}$$

$$\leq \frac{2\left(F(\mathbf{U}^0 \mathbf{V}^{0\top}) - F^\star\right)}{\eta T\big(1 - 2\eta L\big)} + \frac{2\eta L\sigma^2}{1 - 2\eta L},$$

provided that $L = \max\{L_U, L_V\}$, $\eta_v = \frac{p\eta_u}{K} = \frac{p\eta}{K}$, and $\eta \leq \frac{1}{2L}$. This completes the proof.

$\square$

### A.3 Useful Inequalities

We list below a few elementary facts that we used in our analysis, included here only for completeness.

**Lemma 3.** *For any matrices $\mathbf{A} \in \mathbb{R}^{d_1 \times d_2}$ and $\mathbf{B} \in \mathbb{R}^{d_2 \times d_3}$, the Frobenius norm of the product $\mathbf{A}\mathbf{B}$ satisfies*

$$\|\mathbf{A}\mathbf{B}\|_F \leq \|\mathbf{A}\|_2 \|\mathbf{B}\|_F.$$

*where $\|.\|_2$ is the spectral norm.*

**Lemma 4** (Young's inequality)**.** *Let $\boldsymbol{\Theta}, \mathbf{Y} \in \mathbb{R}^{d_1 \times d_2}$ and $\alpha > 0$. Then, the following inequality holds:*

$$\langle \boldsymbol{\Theta}, \mathbf{Y} \rangle \leq \frac{\alpha}{2} \|\boldsymbol{\Theta}\|_F^2 + \frac{1}{2\alpha} \|\mathbf{Y}\|_F^2 \ . \tag{17}$$

**Lemma 5.** *Let $\boldsymbol{\Theta}_i \in \mathbb{R}^{d_1 \times d_2}$ for $i \in 0, \dots, K-1$. Then, the following bound holds:*

$$\left\| \sum_{i=0}^{K-1} \boldsymbol{\Theta}_i \right\|_F^2 \leq K \sum_{i=0}^{K-1} \|\boldsymbol{\Theta}_i\|_F^2 \ . \tag{18}$$

*Proof.* *This inequality follows directly from Jensen's inequality applied to the Frobenius norm.* $\qquad\square$

**Lemma 6.** *Let $\mathbf{D}$ be a diagonal matrix with diagonal entries that are 1 with probability $p$ and 0 with probability $1-p$, and let $\mathbf{A}$ be an arbitrary matrix. Then the expectation of the squared Frobenius norm of the product $\mathbf{D}\mathbf{A}$ is given by*

$$\mathbb{E}_{\mathbf{D}}\left[ \|\mathbf{D}\mathbf{A}\|_F^2 \right] = p\|\mathbf{A}\|_F^2.$$

*Proof.* *The Frobenius norm squared of $\mathbf{D}\mathbf{A}$ is defined as:*

$$\|\mathbf{D}\mathbf{A}\|_F^2 = \sum_{i,j} (\mathbf{D}\mathbf{A})_{ij}^2.$$

*Since $\mathbf{D}$ is diagonal, the product $\mathbf{D}\mathbf{A}$ will zero out all rows of $\mathbf{A}$ where the corresponding diagonal entry in $\mathbf{D}$ is 0. Let $d_i$ represent the $i$-th diagonal entry of $\mathbf{D}$, where each $d_i$ is a Bernoulli random variable with $\mathbb{E}_{\mathbf{D}}[d_i] = p$.*

*Thus, we can express $\|\mathbf{D}\mathbf{A}\|_F^2$ as:*

$$\|\mathbf{D}\mathbf{A}\|_F^2 = \sum_{i=1}^{n} d_i^2 \sum_{j=1}^{m} A_{ij}^2 = \sum_{i=1}^{n} d_i \|\mathbf{A}_{i,\cdot}\|_2^2,$$

*where $\|\mathbf{A}_{i,\cdot}\|_2^2 = \sum_{j=1}^{m} A_{ij}^2$ is the squared norm of the $i$-th row of $\mathbf{A}$.*

*Now, taking the expectation, we have:*

$$\mathbb{E}_{\mathbf{D}}\left[ \|\mathbf{D}\mathbf{A}\|_F^2 \right] = \sum_{i=1}^{n} \mathbb{E}_{\mathbf{D}}[d_i] \|\mathbf{A}_{i,\cdot}\|_2^2 = \sum_{i=1}^{n} p \|\mathbf{A}_{i,\cdot}\|_2^2.$$

*Simplifying, we get:*

$$\mathbb{E}_{\mathbf{D}}\left[ \|\mathbf{D}\mathbf{A}\|_F^2 \right] = p \sum_{i=1}^{n} \|\mathbf{A}_{i,\cdot}\|_2^2 = p\|\mathbf{A}\|_F^2.$$

*Therefore, the expectation of $\|\mathbf{D}\mathbf{A}\|_F^2$ is:*

$$\mathbb{E}_{\mathbf{D}}\left[ \|\mathbf{D}\mathbf{A}\|_F^2 \right] = p\|\mathbf{A}\|_F^2.$$

*This completes the proof.* $\qquad\square$

**Lemma 7.** *Let* $\mathbf{D}$ *be a diagonal* $n \times n$ *matrix where each diagonal entry is independently 1 with probability* $p$ *and 0 with probability* $1 - p$. *Then the expected value of the spectral norm* $\|\mathbf{D}\|_2$ *is given by*

$$\mathbb{E}(\|\mathbf{D}\|_2) = 1 - (1 - p)^n \leq 1.$$

**Proof.** *Since* $\mathbf{D}$ *is diagonal, its spectral norm* $\|\mathbf{D}\|_2$ *is the largest absolute value among its diagonal entries. Therefore,* $\|\mathbf{D}\|_2 = 1$ *if at least one diagonal entry is 1, and* $\|\mathbf{D}\|_2 = 0$ *only if all diagonal entries are 0.*

*Define* $X$ *as the event that all diagonal entries are 0. The probability of this event,* $\Pr(X)$, *is:*

$$\Pr(X) = (1 - p)^n,$$

*since each diagonal entry is 0 independently with probability* $1 - p$.

*Thus, the probability that* $\|\mathbf{D}\|_2 = 1$ *(i.e., the event* $X$ *does not occur) is:*

$$1 - \Pr(X) = 1 - (1 - p)^n.$$

*Therefore, the expected value of* $\|\mathbf{D}\|_2$ *is:*

$$\mathbb{E}(\|\mathbf{D}\|_2) = 1 \cdot (1 - (1 - p)^n) + 0 \cdot (1 - p)^n = 1 - (1 - p)^n \leq 1.$$

*This completes the proof.* $\square$

