# OpenReview forum: "Personalized Federated Learning via Low-Rank Matrix Optimization"
_TMLR — Accepted by TMLR_

### Review · Reviewer_b2RG · 2025-04-14

**Summary Of Contributions:**

In this paper, the authors have looked at the problem of Personalized Federated Learning (pFL) - the goal is to ensure improved model training taking into account relevance and diversity of samples via a low rank formulation of the underlying parameters

**Audience:**

Yes

**Claims And Evidence:**

Yes

**Requested Changes:**

Please differentiate clearly with the existing theoretical work on meta-learning starting with the above one.

**Strengths And Weaknesses:**

The paper is well written but there might be a critical issue.

1. Are the authors aware of the following paper and its follow up works on meta-learning?

https://openreview.net/forum?id=48LtSAkxjiX - this paper studies the exact same problem with the same algorithm (modulo a few minor details) as well. The authors also provide stronger guarantees on convergence to the global optimal in stead of a stationary point.

The same problem formulation, in the context of meta-learning, has been studied already in details (please check the papers that cite the above one as well).

The authors need to differentiate clearly with this line of work.

2. (Minor) x being the notation for parameters looks weird. Why not use \theta?

3. The notation f_i subsumes both the loss personalized for model i and the dataset i. It would be good to clarify this

---

> ### Author Response · Authors · 2025-05-07
>
> Thank you for your time and valuable feedback.
>
> **1. Comparison with meta-learning frameworks**
>
> Thank you for bringing (Thekumparampil et al., 2021) to our attention. While the primary focus of that work is on multi-task learning, we recognize that it can be applied for personalized FL by treating each client's learning problem as a separate task. Indeed, (Collins et al., 2021) considered exactly the same algorithm in the context of personalized FL (and it was cited in (Thekumparampil et al., 2021) as a concurrent and independent work), and we have already compared against them, both contextually in related works and empirically in the numerical experiments sections (see FedRep in the comparisons). We will extend the related work discussion to better highlight the connection with (Thekumparampil et al., 2021) along with other related works that reference it.
>
> The fundamental difference between the representation learning approach in the aforementioned work and our approach lies in the role of low-dimensional structure. Let $\phi(\theta, x^j) \approx y^j$ denote a general machine learning model, where $\theta$ represents the model parameters and $(x^j,y^j)$ denotes the datapoints.
>
> In representation learning, the goal is to find a shared low-dimensional representation of the data features. The model jointly learns a feature extractor $U$ and task-specific parameters $v^i$ such that
>
> $$ \phi'(v^i, U x^i) \approx y^i. $$
>
> Here, $Ux^i$ represents a common low-dimensional embedding of the features shared across tasks and $v^i$ are task specific parameters.
>
> In contrast, our method assumes that the decision variables, i.e., the model parameters $\theta^i$ themselves lie in a low-dimensional subspace. This is modeled as $\theta^i = Uv^i$, so we are looking for
>
> $$ \phi(U v^i, x^j) \approx y^j. $$
>
> This reflects a key modeling difference: rather than learning a shared embedding of the data, we assume that the personalized parameters themselves lie in a low-rank subspace. The two approaches coincide only in the special case of linear predictors where $\phi( \cdot, \cdot) = \langle \cdot, \cdot \rangle$.
>
> Beyond this fundamental difference in modeling, the focus of the two works also differs. In particular, Thekumparampil et al. (2021) focuses on sample complexity and therefore make explicit assumptions on data distributions. Their analysis is restricted to linear predictors and convex loss functions $(\ell)$. It is not possible to establish convergence guarantees to a global solution in the broad nonconvex optimization setting we consider.
>
>
> **2. Notation of parameters**
>
> We will change the notation as suggested, by using $\theta$ for model parameters and $(x,y)$ for datapoints.
>
> **3. $f_i$ notation**
>
> While we believe this is already clear from the definition of $f_i$ in Equation (3), we will include an explicit mention as suggested.

---

> > ### Comment · Reviewer_b2RG · 2025-06-04
> > **Further comments**
> >
> > There is no difference between the two in the linear setting as the authors rightly point out.
> >
> > However, I disagree with the "fundamental" modeling difference being pointed out - in the linear setting there is no difference. However, the authors in (Thekumparampil et al., 2021) also write the problem by projecting the parameters in the low dimensional subspace - see Equation 2 in (Thekumparampil et al., 2021. Algorithmically, there is no difference as well - the only change is to use gradient descent for the sub-steps instead of closed form solutions. The gradient descent step instead of a closed form solution (applicable for linear regression) is not sufficiently novel and has been explored in follow up papers as well - see https://arxiv.org/pdf/2210.03505.pdf for instance.
> >
> > Overall, I feel that given prior work, the significance of the current work is very low.

---

> > > ### Author Response · Authors · 2025-06-13
> > >
> > > Representation learning and matrix factorization are relevant as we already explained in our initial response. These problems become equivalent only under the specific setting of linear measurements, which does not apply to our work, since we focus on neural networks. (Thekumparampil et al., 2021) adopts the linear setting and focuses from a representation learning perspective; as quoted from their introduction: *"The premise is that there is a shared low-dimensional representation $f_U(x) \in R^i$ represented by a task-independent meta-parameter $U$ and a simple linear model of $\langle v_i, f_U(x)\rangle$ can make accurate prediction on the $i$-th task with a task-specific parameter $v_i$."*
> > >
> > > Their main model in Equation 1 also clearly formulates the problem as
> > > $$
> > > \min_{U} \Bigg \\{ \sum_{i} \min_{v_i \in R^r} \sum_{j} \ell( \langle v_i, f_U(x_j^{(i)}) \rangle, y_j^{(i)} ) \Bigg\\}
> > > $$
> > >
> > > Their Equation 2, which you referenced, appears in the context of their Assumption 1 and describes a linear observation model. Nevertheless, throughout the paper, both forms  ($\langle x, Uv \rangle$ and $\langle U^Tx, v \rangle$) appear interchangeably; since they focus on a linear representation and linear measurement regime where representation learning and matrix factorization become equivalent. However, our work lies outside of this regime. Also note that both their model and their algorithm have additional constraints such as orthogonality of $U$, which we do not have.
> > >
> > > We would like to emphasize again that an adaptation of (Thekumparampil et al., 2021) for personalized federated learning is FedRep by (Collins et al., 2021). **We compare against FedRep in our experiments**, and the results show clear performance differences, **highlighting the distinction between the models** outside of the linear setting.
> > >
> > > Regarding the algorithms, they apply the gradient method on $U$ on the function
> > > $$d(U) = \sum_{i} \min_{v_i \in R^r} \sum_{j} \ell (\langle v_i, f_U(x_j^{(i)}) \rangle, y_j^{(i)}).$$
> > > Computing the gradient therefore requires solving the inner sub-problems to obtain the optimal $v_i$s. Additionally, their problem is bi-linear whereas our model is not. We also note that they enforce orthogonality of $U$ after each iteration, a structural constraint that is not present in our formulation.
> > >
> > > In contrast, our algorithm performs gradient steps on both $U$ and $v_i$ (and allows for multiple local updates on $v_i$ **if desired**). When the number of local steps for $v_i$ is large, the method might resemble that of (Thekumparampil et al., 2021) (except orthogonalization steps that they have) where the inner minimization over $v_i$ is solved approximately via gradient descent. However, our results hold even for the case of a **single gradient step**, without requiring convergence to the inner minimum.
> > >
> > > This distinction also clarifies the relation to (Pal et al. 2023) which you referenced in your last comment. While they replace the exact inner minimization with gradient steps, their analysis requires the number of inner updates to be sufficiently large (i.e., $\tilde{\Omega}(\ell)$, where $\ell$ is the outer iteration counter). Our approach, by contrast, remains valid with a single gradient step. Moreover, just like (Thekumparampil et al. 2021), they have additional structural constraints for orthogonality and sparsity, and they also focus on linear representation and linear measurement regime.

---

### Review · Reviewer_eYV6 · 2025-04-15

**Summary Of Contributions:**

The paper presents a well-motivated and technically grounded contribution to the field of Personalized Federated Learning (pFL) through novel low-rank matrix optimization framework. Specifically, the authors propose a new method based on Burer-Monteiro factorization, which assumes that the matrix formed by stacking all clients’ models lies in a low-dimensional subspace.

**Audience:**

Yes

**Broader Impact Concerns:**

-

**Claims And Evidence:**

Yes

**Requested Changes:**

Here are a few comments:

Abstract: The paper mentions a focus on “clustered FL,” where devices are grouped by data similarities. It would be helpful if you provide a short rationale for why low-rank matrix optimization is especially suitable for discovering or leveraging these clusters.

You mention using the Burer–Monteiro factorization technique and that convergence guarantees are examined.  It might be clearer to add one or two lines outlining how the method handles potential numerical instability. If available, highlight any theoretical upper bounds on time/space complexity. That can address practicality concerns right away.

The introduction references the difficulty of balancing local data reliability with other clients’ contributions. This is a strong motivator for your work. However, the text could benefit from a slightly more concrete example. Consider adding a real world example for your motivation

Because you are using a low-rank matrix factorization, the choice of rank can heavily influence both performance and computational complexity. I would like to see a sensitivity test, and comment on how r must be selected. I understand that the authors have tested several values of r, but it can be expanded with sensitivity plots and insights

**Strengths And Weaknesses:**

The experimental design is diverse and relevant, with careful attention to personalized label mappings.

However, the work also has some areas where further clarity or extension would be helpful. First, while the factorization assumption is powerful, the approach may incur high computational or memory overhead when applied to deep models, especially if the entire model is factorized. The paper acknowledges this and suggests partial factorization as future work, so I think the paper is currently okay for TMLR.

Second, although theoretical guarantees are provided, they are somewhat detached from practical choices like the number of local steps K, which appear crucial for empirical success but are not yet reflected in the theory. Lastly, while the baselines are comprehensive, the paper could benefit from deeper ablation studies: such as sensitivity to the rank r or performance under increasing label corruption

---

> ### Author Response · Authors · 2025-05-07
>
> Thank you for your time and valuable feedback.
>
> **1. Clustered FL**
>
> Suppose there are $r$ distinct data distributions. Consider the idealistic setting where the clients have access to the entire distribution. Then, the clients sharing the same data distribution could also share the same optimal solution. It means that the matrix $X = [x_1, \ldots, x_n]$ could be written with only $r$ distinct columns, which implies $\mathrm{rank}(X) \leq r$.
>
> In the more realistic setting where each client observes only a finite sample $\mathcal{M}_i$ (as defined in Equation (3)), the optimal solutions for clients within the same cluster may differ slightly due to sampling variability. However, the problem formulation inherently assumes that these solutions are close to the ideal solution (i.e., those derived from the full data distributions). Otherwise, the formulation fails to capture the true structure of the problem and would not be meaningful to consider. As a result, the solution matrix can be approximated as a low-rank matrix plus small deviations.
>
> **2. Details on Burer–Monteiro factorization**
>
> We did not observe numerical instability issues in our experiments, especially for choices of factorization rank.  We will include guarantees on the time, space, and communication complexity of our algorithms to clarify their practical applicability.
>
> **3. Real-world example**
>
> A notable example is Google’s Gboard, which employs FL to improve next-word prediction (McMahan et al., 2017). Each user’s local data is highly relevant but captures only a narrow segment of language use. Aggregated updates from other users introduce linguistic diversity, but their preferences may not align with an individual’s habits. This naturally motivates the use of pFL, where models are adapted to individual users (or groups) while still benefiting from shared knowledge across the population.
>
> **4. Rank parameter**
>
> The optimal choice of the rank parameter in Burer–Monteiro factorization is a nontrivial task. In general, the rank of the solution is unknown a priori, and even if it were known, choosing the rank below a certain threshold can introduce spurious local minima into the optimization landscape (Waldspurger & Waters, 2018). Notably, recent research has shown that gradient methods applied to Burer–Monteiro factorization problems exhibit an implicit tendency to converge to low-rank (or approximately low-rank) solutions even when the factorization rank overshoots the true rank (Gunasekar et al., 2017; Li et al., 2021). In our experiments, we also observed that choosing $r$ slightly larger than the actual number of underlying data distributions (i.e., clusters) generally gives a better performance. However, setting $r$ too large can lead to computational inefficiencies, so it is important to balance model expressiveness with efficiency.
>
> As a practical guideline, if the number of underlying data distributions is known, we suggest choosing $r$ to be two to three times that number. Otherwise, start with a small $r$ and retrain the model with progressively larger values (e.g., doubling it each time) until performance saturates (or degrades) or meets the desired level.
>
> **References**
> - Gunasekar et al., Implicit regularization in matrix factorization, 2017.
> - Li et al., Towards resolving the implicit bias of Gradient Descent for matrix factorization: Greedy low-rank learning, 2021.
> - Waldspurger & Waters, Rank optimality for the Burer-Monteiro factorization, 2018.
> - McMahan & Ramage, Federated learning: Collaborative machine learning without centralized training data. Google AI Blog, 2017. https://ai.googleblog.com/2017/04/federated-learning-collaborative.html

---

### Review · Reviewer_BrW6 · 2025-05-05

**Summary Of Contributions:**

The paper develops a personalized federated learning algorithm. Specifically, the paper focuses on the clustered FL setting where clients are grouped into clusters based on similarities of data distributions without having prior knowledge of cluster memberships. However, instead of directly clustering clients, the authors provide a new formulation based on low-rank matrix factorization, where the $n$ clients' models are assumed to lie in an $r$ dimensional space. The authors then proceed to develop an algorithm where factor models based federated learning is carried out across clients. Particularly, the local model at a client can be described as product between a shared low-rank matrix across clients, and a personal part corresponding to the specific client. The authors also provide a convergence analysis of their algorithm. Experiments with neural networks are also provided.

**Audience:**

Yes

**Broader Impact Concerns:**

I do not have any broader impact concerns.

**Claims And Evidence:**

Yes

**Requested Changes:**

Please make the changes required to address the Weaknesses mentioned above.

**Strengths And Weaknesses:**

**Strengths**
The proposed framework for personalized FL based on low-rank factorization is new and interesting. The paper is generally well-written, the work is grounded in theory and experiments demonstrate the performance of the proposed algorithm in practice.

**Weaknesses**
Many relevant prior works are not discussed/compared with. For example, multiple prior works have addressed various notions of `noisy' labels in FL scenarios [1], [2]. With respect to baselines, why not consider FedHM for comparison? I understand that it is not specifically for personalization, but it would be good to see comparison results for the global model. In relation to that, the work of [3] is also quite relevant for comparison.

Additionally, have the authors tried an FL type update on $U$? I mean each client updates $U$ based on local information and then server averages the updates from the clients?

Furthermore, I am not sure why the baselines considered in the experiments are so old. Why not compare with Mishchenko et al. (2023); Pillutla et al. (2022), which have been cited in the paper?

[1] Bao, W., Wang, H., Wu, J., & He, J. (2023). Optimizing the collaboration structure in cross-silo federated learning. In ICML.
[2] Prakash, S., Sima, J., Pan, C., Chien, E., & Milenkovic, O. (2023). Federated classification in hyperbolic spaces via secure aggregation of convex hulls. In ICLR.
[3] Niu, Y., Prakash, S., Kundu, S., Lee, S., & Avestimehr, S. (2023). Overcoming resource constraints in federated learning: Large models can be trained with only weak clients. Transactions on Machine Learning Research.

---

> ### Author Response · Authors · 2025-05-07
>
> Thank you for your time and valuable feedback.
>
> **1. References**
>
> Thank you for pointing us to these references. We will revise the related work section in light of your suggestions.
>
> **2. FL updates on $U$**
>
> Do you mean using multiple local steps for the $U$ update before communication? We have not tried it, but it is an excellent suggestion. We will add experiments to evaluate this idea.
>
> **3. Baselines**
>
> We will include the baselines you suggested in our experimental evaluation.

---

> ### Comment · Reviewer_BrW6 · 2025-06-04
> **Follow-up Comments**
>
> Thanks to the authors for addressing the comments to some extent. However, they have missed the following from my previous set of reviews:
>
> 1. While [1] and [2] have been now referenced in the revised paper, the relevant context of `noisy labels in FL scenarios' is missing in the paper. Also, [2] is published in TMLR https://openreview.net/forum?id=umggDfMHha (not in ICLR as I had mistakenly mentioned in my previous review), please provide full citation.
>
> 2. A comparison with FedHM or [3] is missing. Please provide the comparison results or justify why FedHM or [3] could not serve as baselines.
>
> 3. The results with local iterations for $U$ is missing.

---

> ### Author Response · Authors · 2025-06-13
>
> As you have already called it in your first response, FedHM and PriSM are not for personalized federated learning. They are specifically designed for system heterogeneity (i.e., resource-constrained settings where the bottleneck is varying computational resources among the clients, not different data distributions). Therefore, these algorithms aim to train a single global model in a collaborative manner. It is unclear how these algorithms should be evaluated in a personalized federated learning setting where test distributions differ across clients. Consider, for example, when the labels are permuted.
>
> Additionally, we believe the reference you linked ("Federated Classification in Hyperbolic Spaces via Secure Aggregation of Convex Hulls") may have been included in error, as it focuses on convex SVM classifiers in hyperbolic space and appears to be unrelated to our paper. We would appreciate further clarification if there is a specific connection to our work.
>
> Example 1 in our paper explicitly calls the noisy label setting a special case of personalized federated learning.
>
> We will perform additional experiments with local steps on $U$.

---

> > ### Comment · Reviewer_BrW6 · 2025-06-16
> > **Follow-up Comments**
> >
> > Thanks to the authors for their response.
> >
> > Thanks for the clarification regarding FedHM and PriSM as baselines, I appreciate it.
> >
> > Regarding [2], please check out the discussion after Figure 2 in that paper, it considers a very similar flipped label scenario as in your example 1. While [2] generally considers the hyperbolic classification problem, the solution for addressing `label switching' (see Sections 5.4-5.6), is generic and can be used for Euclidean FL as well. Personalized FL can be applied on top of the grouping result of [2] (Section 5.6).
> >
> > Finally, please also provide results with local iterations of $U$.

---

### Author Response · Authors · 2025-05-22
**General response**

Dear Reviewers,
Thank you for your time and valuable feedback.

**Below, we highlight the changes made in the revised draft.**

**1.** We have extended our comparisons to include **four additional algorithms**, covering both recent personalized methods and conventional baselines.

**2.** We have added **sensitivity plots** for $\texttt{pFL}^{\texttt{MF}}$ with respect to the choice of rank. These include both accuracy-versus-rank and runtime-versus-rank plots.

**3.** We have incorporated **discussions of the suggested papers** and clarified how our approach differs from them.

**4.** **Notation has been updated** from $x$ to $\theta$, and we have added clarification regarding the dependence of local objectives on client-specific data.

All newly added content in the revised paper is highlighted in **blue** for clarity.

---

### Decision · Action_Editor_Zyy9 · 2025-07-12

**Recommendation:** Accept with minor revision

**Additional Comments:**

The reviewers have mixed reactions to this paper - with major objection mostly to the novelty of the paper given a prior work on shared representation learning. In my reading, the formulation and methods of current paper differs from the mentioned prior literature. In any case, the amount of novelty is a subjective measure and according to TMLR guiding principles, limited novelty is not a deciding factor in rejecting submissions.

It will nonetheless be useful to compare and discuss the algorithm of ``Pal et al., Sample-Efficient Personalization: Modeling User Parameters as Low Rank Plus Sparse Components''in the final version, in addition to the comparisons the authors already made.

**Audience:**

Yes

**Audience Explanation:**

Federated learning is quite a popular topic - and data heterogeneity in federated learning is a major and central challenge. The new methods will be useful for federated learning practitioners.

**Claims And Evidence:**

Yes

**Claims Explanation:**

The paper proposes a new method for personalized federated learning based on low rank matrix factorization. The authors have shown convergence results for their algorithm under realistic/usual assumptions. The results seem to be correct; and I found the experimental results to be sufficient.